# Ambiguous Images With Human Judgments for Robust Visual Event Classification

**Kate Sanders**    **Reno Kriz**    **Anqi Liu**    **Benjamin Van Durme**
Johns Hopkins University
Human Language Technology Center of Excellence
{ksande25, rkriz1, aliu74, vandurme}@jhu.edu

## Abstract

Contemporary vision benchmarks predominantly consider tasks on which humans can achieve near-perfect performance. However, humans are frequently presented with visual data that they cannot classify with 100% certainty, and models trained on standard vision benchmarks achieve low performance when evaluated on this data. To address this issue, we introduce a procedure for creating datasets of ambiguous images and use it to produce SQUID-E ("Squidy"), a collection of noisy images extracted from videos. All images are annotated with ground truth values and a test set is annotated with human uncertainty judgments. We use this dataset to characterize human uncertainty in vision tasks and evaluate existing visual event classification models. Experimental results suggest that existing vision models are not sufficiently equipped to provide meaningful outputs for ambiguous images and that datasets of this nature can be used to assess and improve such models through model training and direct evaluation of model calibration. These findings motivate large-scale ambiguous dataset creation and further research focusing on noisy visual data.[1]

## 1 Introduction

When making decisions, the human brain uses perceptual uncertainty judgments to account for missing visual information and other noise [22, 2, 26]. For instance, when humans enter a new environment, they must quickly gauge what events are taking place in it using limited sensory input [53, 80, 79]. However, this practice is not reflected in most vision models. Robustness to out-of-domain or otherwise noisy data has been an area of focus within the computer vision community in recent years [24, 4] with various studies showing model limitations in this regard. However, little work has been done on classifying images that humans also struggle to classify accurately. This lack of emphasis on ambiguous data can cause poor model performance on noisy images that require human uncertainty quantification.

Due to the temporal nature of events and their semantic complexity, visual event classification invites significant data-driven ambiguity. A common task in visual event classification is situation recognition, where a model must identify the verb and corresponding semantic roles (e.g. subject, object, place, reason, etc.) depicted in an image to characterize the event taking place [78]. In a typical framework, a situation recognition model first classifies the verb depicted in the image using an action recognition network and then, given that predicted verb, recurrently identifies semantic roles associated with it shown in the image [52]. This paradigm lends itself particularly poorly to ambiguous data, as the output of the model depends heavily on correctly identifying the verb depicted using only raw pixels as input. If even humans cannot identify the verb taking place with certainty,

---

[1]Dataset and code are available at https://katesanders9.github.io/ambiguous-images.

36th Conference on Neural Information Processing Systems (NeurIPS 2022) Track on Datasets and Benchmarks.

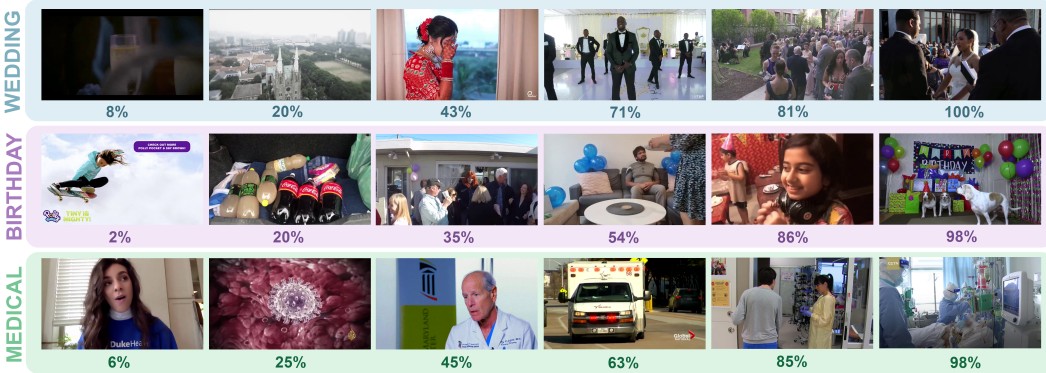

Figure 1: Images from SQUID-E and their corresponding human judgment scores. Each judgment score shows the mean human-elicited likelihood of the image depicting the event listed on the left in that row (wedding, birthday party, or medical procedure).

then it follows that these models will rarely output meaningful information even when they accurately recognize important event-centric attributes in the data.

Ignoring ambiguity in data can lead to significant consequences in downstream applications. For example, autonomous agents that collaborate with humans in tasks like manufacturing must accurately assess this kind of ambiguous event-based data (such as visual input showing what its human partner is doing) to make behavioral decisions [76] and ensure user safety [6, 41]. This requires an ability to produce reliable outputs under perceptual data-driven uncertainty. In some scenarios, a model's calibration scores may additionally need to resemble judgments made by humans to imitate human behavior and allow for better model interpretation [67].

In this paper, we consider the question of how to construct a dataset of noisy images for uncertainty-aware situation recognition. We propose a method for collecting ambiguous images from internet videos and assigning them human judgment scores. This data collection process mitigates the reporting bias found in existing image datasets to model the distribution of visual input experienced by humans. Using this method, we present the first dataset of intentionally ambiguous event-centric images: SQUID-E, or the Scenes with Quantitative Uncertainty Information Dataset for Events. To our knowledge this is also the first event-centric image dataset that uses quantitative human uncertainty judgments as labels. We show in experiments that these images and labels can be used to train robust classification models, assess model accuracy on different distributions of ambiguous data, additionally directly assess model calibration techniques. Sample images and human judgments from SQUID-E are shown in Figure 1.

In summary, we make the following contributions:

1. We introduce a novel method for generating visual uncertainty datasets consisting of noisy images scraped from videos and corresponding human uncertainty judgments.

2. We use this process to construct SQUID-E which consists of 12,000 images with ambiguous contexts, corresponding context labels, and 10,800 human uncertainty judgments for a test set of 1,800 of these images.

3. We demonstrate the applicability of ambiguous image datasets through experiments: We show that training on ambiguous datasets may result in up to a 9-point accuracy improvement on other ambiguous data, existing situation recognition models do not necessarily produce meaningful outputs for ambiguous data, and human uncertainty scores for noisy data can be used to evaluate model calibration approaches.

## 2 Related Work

### 2.1 Collecting Ambiguous Data in Computer Vision

While uncertainty in machine learning has been widely studied through lines of work such as model calibration [1], the practice of using collections of human uncertainty judgments in such efforts is

rare. Previous work on collecting information beyond a single label are largely motivated by the issues of training and evaluating models using the standard "clean" dataset. For example, Beyer et al. [11] explore the issue of model overfitting on standard labeling paradigms such as that used to construct ImageNet [21] by using soft human-annotated label distributions, exploring how even "high-certainty" data can elicit variance in human responses. Along this line of work, Peterson et al. [51] construct a dataset of CIFAR-10 [36] images labeled with distributions of human judgments. Other work [63, 19, 70] considers frameworks for learning from fuzzy human labels given possibly ambiguous data. Our dataset differs from these papers in that none of the listed datasets include images that are intentionally ambiguous, or depict more than a single object.

Furthermore, the human labels used in previous datasets do not include individual human uncertainty judgments. For example, Misra et al. [42] explore the relationship between the explicit contents of an image and the corresponding semantic components that humans label, characterizing the reporting bias of humans, while we solicit quantitative uncertainty judgments pertaining to the image as a whole and its relationship to event classifications instead. Additionally, we provide possible explanations for how humans make these judgments. Another notable research effort is Chen et al.'s work [15] which introduces a dataset of human entailment judgments for the Uncertain Natural Language Inference task in which a model must directly predict these human uncertainty scores. Their annotation process is similar to ours, but is used for text annotation as opposed to images.

## 2.2  Assessing Human Uncertainty in Ambiguous Data

Works in cognitive science provide a natural motivation for the machine learning community to explore the usage of ambiguous data or uncertainty quantification methods when data are collected from humans. Classic works have introduced a variety of notable ideas including theoretical uncertainty taxonomies [12], the impact of implicit bias and context on judgments [66, 64, 46], and strategies humans use to produce judgments. [68, 10]. Many papers consider how human uncertainty scores align with probability theory [55, 2, 71]. Ma et al. identify the performance gain achieved by organisms who incorporate uncertainty measures through visual processing, etc. into their decision-making [40, 23, 34, 22] and perceptual organization [81]. Our work differs from these papers in that we approach this concept from a machine learning perspective.

Work concerned with ambiguity in data in machine learning frequently quantifies it by assessing the aleatoric, or data-driven uncertainty in systems [33]. In the vision community, aleatoric uncertainty is typically considered in the context of medical image processing, where model calibration is necessary to employ agents in high-stakes applications. A popular method of quantifying aleatoric uncertainty in medical imaging is data augmentation [5, 62], but authors such as Beluch et al. [9] and Reinhold et al. [56] use alternate techniques such as ensembling and dropout network layers. Nado et al. [47] produce a system for benchmarking such methods, but does not consider using human uncertainty scores to assess models. To our knowledge no work exists that considers ambiguity in semantically complex images, such as ones that depict events.

Various work has also considered aleatoric uncertainty estimation from a more theoretical perspective. For example, works have considered how aleatoric uncertainty can be measured by estimating the parameters of a Gaussian distribution by maximizing the log likelihood [65, 49, 37, 33]. Other work considers how outputs produced using this method can be further improved in the case of regression [45, 31, 44, 48] and categorical classification [45, 31, 44, 48]. However, these works usually only consider the aleatoric uncertainty estimation problem in the existing data, but do not directly consider datasets with human uncertainty judgement. In this paper, we consider different sources of uncertainty in the context of human uncertainty judgments. We also investigate how these judgments can be used to train and evaluate models in situation recognition applications.

## 2.3  Situation Recognition and Verb Prediction

Initially, event-centric image classification was primarily constrained to simple tasks like pose estimation. Early forays into more sophisticated event classification proposed organizing images through frameworks that draw from linguistic event semantics [28, 58]. This proposal was built on by Yatskar et al. [78] who introduced a FrameNet-based ontology for event classification. Many papers consider novel model architectures and extensions based on this work, some exploring multi-modal extensions [38, 73], and others implementing bounding box grounding [52]. Wei et al. [74] and

Cho et al. [16] introduce new models that depart from the typical two-stage classification pipeline to better model event attribute relationships. Cho et al. [17] incorporate transformers in the original architecture, Sadhu et al. [60] apply the framework to video understanding, and Dehkordi et al. [20] alternatively use a CNN ensembling method. In all of these approaches, it is assumed that the necessary elements to identify the event are clearly depicted in the image, and it is not explored how these models perform when presented with ambiguous data. This is what we explore in this paper.

It is typical for situation recognition models to first predict the verb depicted in a given image and then predict the various semantic roles associated with that verb [52, 17, 16]. Given that the semantic role classification depends on accurate verb classification, it is critical for systems to retrieve reliable information regarding the possible verbs depicted in an input. Therefore, in this paper we focus on the verb prediction modules of these systems. However, these verb prediction modules typically do not consider uncertainty, or image-driven ambiguity. In our work, we consider uncertainty quantification for ambiguous data and focus on techniques that account for aleatoric uncertainty.

## 3 Dataset Construction

In this section we detail our ambiguous dataset construction approach which we used to develop SQUID-E. Our process consists of (1) scraping YouTube for videos of specific events, (2) extracting visually distinct images from these videos, (3) identifying careful annotators to provide human judgments, and (4) executing the full annotation task using images collected in steps (1) and (2) to produce a set of corresponding human uncertainty judgments.

### 3.1 Image Collection

**Considering Reporting Bias** The distribution of still images found in publications, internet image libraries, or other corpora is influenced by reporting bias [27]: It may not accurately reflect the distribution of visual data humans perceive in real life because images in these corpora are, typically, intentionally selected to maximize saliency. Therefore, most datasets contain images that are generally easy for humans to classify with high certainty. To produce a dataset of noisier, more ambiguous visual data, we extracted images from videos. While videos still suffer from this reporting bias, the video corpora curation processes typically consider the comprehensive contents of an entire video rather than the individual frames that comprise it, resulting in large collections of noisier images that can still be easily classified based on content.

**Video Collection and Image Extraction** We considered 20 event types, covering topics such as various social activities, sports, and natural disasters. We intentionally selected event types that typically take place over relatively long time frames, allowing for a wide variety of images that can belong to these event types. To populate the dataset, we first scraped YouTube for videos that fall into one of these twenty event types using YouTube's search algorithm. Search queries involved the name of the event type and related keywords, and were made in multiple languages for each event type. In SQUID-E, we additionally included a selection of videos from the Extended UCF Crime dataset [50]. Retrieved videos were manually checked and removed if they were not relevant or did not contain sufficiently diverse visual content. This process resulted in a collection of 100 videos per event category. Six frames were extracted from each video using a combination of frame sampling and clustering to produce a collection of visually diverse images from each video. Further image collection details are included in Appendix A.1.

### 3.2 Human Uncertainty Judgment Solicitation

**Annotation Setup** Annotations were collected for a set of six event types using Amazon Mechanical Turk. Annotators were provided with an event prompt and six images from the dataset. They were then told to rate their confidence that each image belonged to a video depicting the prompted event type using sliding bars ranging from 0% to 100%. Annotators were provided with three example images of each event type and were shown guidelines explaining how to rate their uncertainty on a numerical scale. Notably, annotators were instructed to only rate an image 0% if the image contained a set of visual attributes that necessarily could not appear together alongside the target event type, and rate an image 100% if the image contained a set of attributes that, together, could only belong alongside the target event type. An image should be rated 50% if the annotator felt there was an

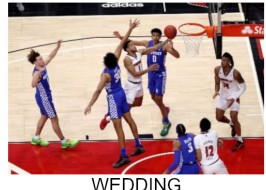 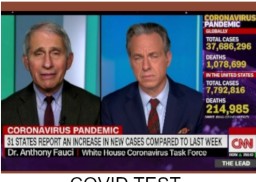 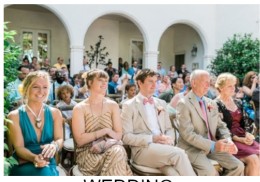 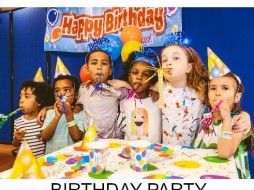

| WEDDING | COVID TEST | WEDDING | BIRTHDAY PARTY |
| 0% | 15% - 35% | 65% - 85% | 100% |

Figure 2: Example images and corresponding ratings given a target event type. From left to right: (1) The visual attributes in this image virtually never coincide with a wedding event. (2) The image contains attributes closely related to COVID tests, but does not contain attributes that would necessarily appear in a COVID test setting. (3) This image has many attributes that often appear in a wedding, but these attributes also could appear in a related event type. (4) Most of the attributes in this image are uniquely characteristic of a birthday party.

equal likelihood that the image belonged to a video of the target event type and that the image did not. Examples of images and appropriate ratings given a target event type are shown in Figure 2. The full instructions given to annotators and a screenshot of the annotation setup can be found in Appendix B.

**Selecting Annotators**    A small pilot was first run to identify high-quality annotators. As discussed in the following section, high disagreement on individual image scores is an intrinsic aspect of the task. Therefore, a purely numerical metric such as mean squared error against a ground truth vector could not be used to identify high-quality annotators. Instead, for each set of images, a rubric identifying possible annotator "pitfalls" was established and used to identify lower-quality annotators who either did not read the instructions or did not apply the instructions correctly when evaluating images. Such pitfalls included heuristics such as rating completely black images above 5%, rating images with text clearly stating the event type below 90% or above 10% (depending on the target event and text), etc. This process produced a set of ten to twenty high-quality annotators per task.

**Task Variants**    To identify whether the other five images on screen influence annotator uncertainty scores for an image, we ran two versions of the annotation task. These two variants present annotators with different sets of images at a time. In Variant A, each of the six images that appeared on screen belonged to the same event type, but were sampled from different videos. In Variant B, given a target event type, three frames on screen belonged to videos depicting that target event type, and the other three frames belonged to videos of the event type most semantically similar to the target event type (e.g., three frames from the "birthday" event type and three frames from the "wedding" event type, or three frames from the "parade" event type and three frames from the "protest" event type). The semantic similarity of events was calculated using FrameNet templates. No annotator participated in both task variants after the pilots were completed.

## 4    Ambiguity of Data

### 4.1    Inter-Annotator Agreement

We explicitly aim to quantify the data-driven noise in images through the dataset's numerical human judgments. However, the set of collected human judgments has its own inter-annotator variance. In SQUID-E, the Spearman correlation among annotators is 0.676 for Variant A, 0.631 for Variant B, and 0.673 across both variants, indicating that there was not a substantial overall difference in annotator behavior between the two versions of the task, although annotators were slightly more confident when annotating for variant B as shown in Figure 3. Alternative metrics for assessing annotation agreement are considered in Appendix C.

### 4.2    Intra-Annotator Agreement

While most of the analysis in this section focuses on inter-annotator variation, it is also important to consider how a person may annotate the same data differently across multiple samples, and how this intra-annotator variance may influence the annotations of the dataset. To assess this factor, the six annotators who labeled the most images in the original annotation task were given the task again for a random subset of the images they had already annotated. This study was conducted five months

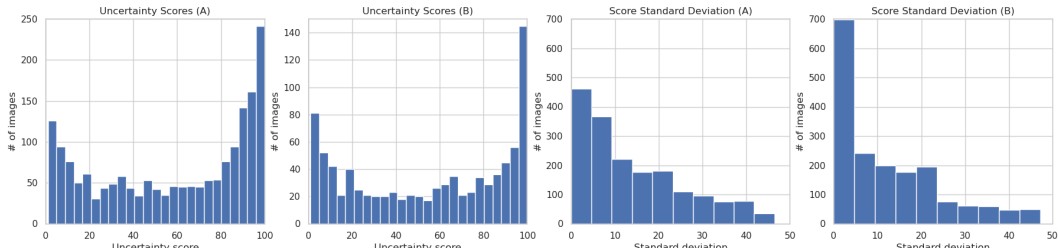

Figure 3: Histograms illustrating the distribution of mean human uncertainty judgments and standard deviation scores among annotations for each image in task variants A and B. Annotators were slightly more confident in task variant B: For variant A, 16% of images were rated as "high certainty", or received a mean certainty score at or above 95%, whereas 19% of images were rated as high certainty when annotated in variant B.

after the original annotations had been collected. Five of the six annotators accepted the task (2 from variant A and 3 from variant B) and annotated 60 images each. The Spearman correlation between an annotators' original scores and their second set of scores was calculated, and the mean correlation was 0.788. This result suggests that some of the variance in the scores between different annotators is likely due to irreducible variance that would occur even if the two annotators were exactly the same. However, the intra-annotator spearman correlation is still significantly higher than the inter-annotator spearman correlation (0.788 vs. 0.673), indicating that there are additional factors that contribute to different humans rating images differently. We explore these factors below.

## 4.3 Sources of Annotator Disagreement

Why do some images produce human judgments with high variance, while others elicit general agreement? Here, we compile possible explanations that illustrate more general human uncertainty quantification trends. Examples of these categories are provided in Figure 4.

Based on an analysis of the human judgments collected for this dataset, the primary sources of inter-annotator variance likely include differences in the following:

- **Visual attention.** A person's visual attention can affect their perceptual input and uncertainty calculations [14, 13, 57, 43, 42]. Some studies suggest that visual attention may even directly cause conservative bias in perception [54]. We hypothesize that this phenomenon affected the human judgments in our task, since the images in SQUID-E can often be classified as multiple event types depending on where an annotator's visual attention is focused.

- **Background knowledge.** Many images require an annotator to hold specific knowledge to classify them accurately, and so people may annotate these images differently depending on their personal knowledge bases. Necessary background knowledge is often cultural, or otherwise related to current events or history. This source of disagreement highlights the importance of formulating tasks and datasets such that they include diverse data and are annotated by diverse annotators [39].

- **Uncertainty quantification strategies.** Prior work explores various sources of human probability estimation error and noise, indicating that one notable factor is that the way humans calculate probability is inherently imperfect. [68, 32, 25] These studies detail heuristics and psychological biases that influence human judgments that are not necessarily caused by external factors such as input or contextual knowledge. We hypothesize that this type of internal factor, divorced from visual input and knowledge bases, may affect annotator score discrepancy.

In addition to these sources, it is highly probable that some of the annotation variance can be attributed to annotator carelessness. In other cases, the underlying cause of this variance may be due to a combination of the sources listed above. The complexity of the factors affecting annotator uncertainty scores illustrates the nuance of human uncertainty judgments and indicates that more research may need to be conducted in this field to better understand in what way humans differ in their processing of uncertain sensory input.

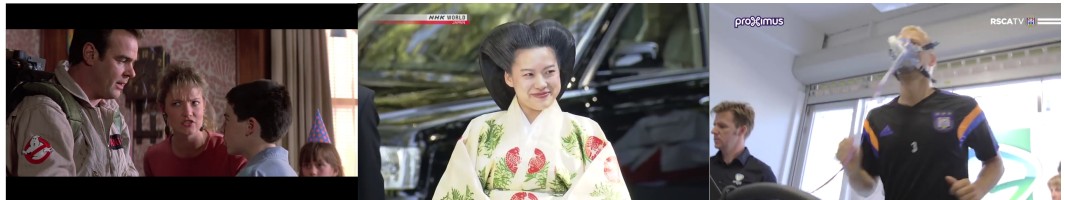

Figure 4: Examples of images that may elicit annotator disagreement. Left to right: (1) **Visual attention**: An image of a birthday party that may be mislabeled due to attention differences, since the key event-based attribute is not centered in the frame. (Background knowledge of the source film may also affect values for this image). (2) **Background knowledge**: An image frame from Princess Ayako's wedding, which was a prominent current event in Japan. Annotators who saw footage from the wedding or are familiar with Japanese wedding traditions will likely have higher certainty scores for this data. (3) **Uncertainty quantification**: An image from a pulmonary function test that is ambiguous enough that it may receive different uncertainty scores from annotators with different risk tolerances - some annotators may naturally give a lower confidence score despite having the same background knowledge.

## 5    Experiments

We run three experiments using SQUID-E to demonstrate the uses of ambiguous image datasets in training and evaluating models: (1) We show how training models on ambiguous images can improve their accuracy when classifying other ambiguous images by comparing models trained on "high-certainty" data with a model trained on SQUID-E. (2) We illustrate how SQUID-E can be used to assess existing situation models' performance on varying degrees of noisy data by evaluating state-of-the-art models on a subset of SQUID-E and partitioning their performance on SQUID-E's human labels. (3) We explore how SQUID-E can be used to directly evaluate model calibration techniques by comparing model confidence scores to SQUID-E's human labels. Additional details regarding experiments are included in Appendix D. We use mean human annotation scores as human labels for these experiments, which is compared to other aggregation approaches in Appendix D.3.

### 5.1    Training on SQUID-E for Event Classification

**Models**    We compare the accuracy of models trained on standard, "high-certainty" data and models trained on ambiguous images. We train one ResNet-based model using images from SQUID-E (RN+ES), and a set of ResNet-based models using images from standard, high-certainty image datasets (Visual Genome [35], Crowd Activity [72], USED [3], WIDER [77], and UCLA Protest Images [75]): RN+SD is trained on the images with no augmentation techniques, RN+PA is trained with photometric augmentation filters, RN+GA is trained with geometric augmentation filters, RN+NM is trained with noise injection and masking, RN+AU is trained with a combination of the augmentation filters listed above, and RN+AM is trained with the AugMix augmentation method [30].

**Task**    We consider a four-way classification task using birthday party, wedding, parade, and protest images (with their respective event names as labels), since these are the four event types that are both represented in existing high-certainty image datasets and have human labels in SQUID-E. We train each model on these four event types using the same number of images from the two datsets. We run this experiment across 10 seeds and report mean accuracy and standard deviation in Table 1.

**Results**    The results in Table 1 suggest that while data augmentation can improve model results on ambiguous data, it is not necessarily as effective as training on ambiguous data. The results indicate that this is the case even for images with relatively high certainty annotations. However, RN+ES having high accuracy scores for the lower certainty bins seems to indicate that it is poorly calibrated compared to the other approaches. While this may just be poor calibration, we hypothesize that this result is at least partially caused by the small set of possible classification labels in the experiment. In our annotation task, the human annotators had to account for the full range of possible event types that could occur in a video, but in this experiment the models only select between four event types.

Table 1: Accuracy of verb classifiers on SQUID-E (bins, Avg. Acc) and standard data (SD Acc) when trained on standard data (RN+SD), augmented standard data (RN+PA, RN+GA, RN+MA, RN+AU, RN+AM), and SQUID-E (RN+ES). Results are partitioned into bins based on human-judged ambiguity. Best results for average accuracy are listed in bold. Details regarding data augmentation techniques are located in appendix D. While augmentation methods improve model performance on uncertain data, they do not achieve the accuracy scores of a model trained on ambiguous data.

| Model | 0-20% | 20-40% | 40-60% | 60-80% | 80-100% | Avg. Acc | SD Acc. |
|---|---|---|---|---|---|---|---|
| RN+SD | $.34 \pm .07$ | $.47 \pm .05$ | $.53 \pm .03$ | $.69 \pm .02$ | $.75 \pm .02$ | $.61 \pm .02$ | $\mathbf{.93 \pm .00}$ |
| RN+PA | $.27 \pm .03$ | $.32 \pm .02$ | $.47 \pm .05$ | $.65 \pm .02$ | $.74 \pm .02$ | $.57 \pm .02$ | $.90 \pm .01$ |
| RN+GA | $.38 \pm .05$ | $.37 \pm .03$ | $.57 \pm .03$ | $.63 \pm .03$ | $.78 \pm .01$ | $.64 \pm .01$ | $.89 \pm .01$ |
| RN+NM | $.28 \pm .05$ | $.45 \pm .03$ | $.61 \pm .02$ | $.69 \pm .02$ | $.81 \pm .02$ | $.64 \pm .02$ | $.91 \pm .01$ |
| RN+AU | $.35 \pm .03$ | $.37 \pm .02$ | $.57 \pm .03$ | $.61 \pm .02$ | $.80 \pm .01$ | $.64 \pm .01$ | $.87 \pm .01$ |
| RN+AM | $.31 \pm .05$ | $.40 \pm .08$ | $.53 \pm .03$ | $.69 \pm .02$ | $.77 \pm .03$ | $.60 \pm .02$ | $\mathbf{.93 \pm .00}$ |
| RN+ES | $.51 \pm .04$ | $.49 \pm .03$ | $.70 \pm .04$ | $.67 \pm .02$ | $.81 \pm .02$ | $\mathbf{.70 \pm .01}$ | $.71 \pm .02$ |

Table 2: Accuracy of situation recognition models on SQUID-E. Results are partitioned on human judgments of the data. Results are partitioned into bins based on human-judged ambiguity. Results on the original SWiG dataset are reported under the column titled "SD (Standard Data) Acc.". Accuracy of the top scoring verb as well as the top 10 scoring verbs are reported (listed as "Top 1" and "Top 10" respectively). Best results for average accuracy are listed in bold. Results show how model performance is affected by different levels of uncertainty in the validation data.

| Model | 0-20% | 20-40% | 40-60% | 60-80% | 80-100% | Avg. Acc | SD Acc. |
|---|---|---|---|---|---|---|---|
| Situation Recognition Model Verb Accuracy (Top 1) | | | | | | | |
| JSL | .00 | .07 | .17 | .22 | .52 | .35 | .40 |
| GSRTR | .02 | .09 | .22 | .25 | .59 | **.41** | .41 |
| CoFormer | .02 | .13 | .22 | .23 | .58 | .40 | **.45** |
| Situation Recognition Model Verb Accuracy (Top 10) | | | | | | | |
| JSL | .11 | .43 | .49 | .72 | .86 | .66 | - |
| GSRTR | .11 | .55 | .54 | .77 | .88 | .70 | - |
| CoFormer | .09 | .42 | .58 | .82 | .91 | .70 | - |

## 5.2 Evaluating Verb Prediction Models

**Models**   To assess contemporary situation recognition models on ambiguous data, we evaluate the verb prediction modules of three high-performing models on the SQUID-E test set: JSL (Platt et al. [52]), GSRTR (Cho et al. [17]), and CoFormer (Cho et al. [16]). These models were selected because of their varied architectures and strong performance on existing benchmarks, and because they have publicly available model weights, ensuring accurate comparisons. JSL uses a ResNet-based verb predictor, GSRTR uses a transformer encoder for verb prediction, and CoFormer uses two transformers for verb prediction (a "glance" transformer and a "gaze" transformer). All three models are trained on the SWiG dataset [52] and can classify 504 distinct verbs in images. We specifically assess the models' verb prediction modules because verb prediction makes up the foundational task of situation recognition and aligns with the course-grained event information we ask annotators to judge in the data collection process. A more detailed explanation is included in appendix D.2.

**Task** We evaluate each verb classifier on "parade" and "protest" images in SQUID-E, because these are the two event types in SQUID-E that have human labels and also exist within the ImSitu ontology. We consider the top scoring verb from each model as well as the top 10 scoring verbs. Results are partitioned on the images' human uncertainty scores, and are reported in Table 2 along with top-1 verb prediction performance on SWiG as reported in the models' respective papers.

**Results**   This experiment demonstrates how we can use SQUID-E to characterize model performance on noisy data. Using the bin partitions in Table 2, we are able to identify average model performance at different levels of ambiguity. By comparing top-1 accuracy to top-10 accuracy, we are able to identify how much of the accuracy drop between bins is due to fine-tuning compared

Table 3: Evaluation of uncertainty quantification methods using accuracy on standard data (SD Acc), accuracy on SQUID-E (SE Acc), MSE against human judgments (HUJ MSE), and expected calibration error (ECE) (calculated using ground truth labels). Best results are listed in bold. (BL - Baseline, MC - Monte-Carlo, LS - Label Smoothing, BM - Belief Matching, FL - Focal Loss, RS - Relaxed Softmax). Results for the SD-trained model in particular shows that uncertainty quantification techniques can produce models that better align with human uncertainty scores while also improving the expected calibration error, indicating that human judgments can be used for calibration evaluation.

| | Trained on SD | | | | Trained on SQUID-E | | | |
| | SD Acc | SE Acc | HUJ MSE | ECE | SD Acc | SE Acc | HUJ MSE | ECE |
| --- | --- | --- | --- | --- | --- | --- | --- | --- |
| BL | $.92 \pm .02$ | $.69 \pm .02$ | $.15 \pm .05$ | $.58 \pm .02$ | $.76 \pm .03$ | $.73 \pm .04$ | $.16 \pm .04$ | $.61 \pm .05$ |
| MC | $.92 \pm .02$ | $.69 \pm .02$ | $.14 \pm .05$ | $.57 \pm .03$ | $.76 \pm .03$ | $.72 \pm .05$ | $.16 \pm .04$ | $.57 \pm .04$ |
| LS | $.92 \pm .02$ | $.69 \pm .02$ | $.12 \pm .03$ | $.46 \pm .02$ | $.77 \pm .03$ | $.73 \pm .04$ | $.14 \pm .03$ | $.52 \pm .04$ |
| BM | $.92 \pm .02$ | $.69 \pm .02$ | $.14 \pm .05$ | $.55 \pm .02$ | $.76 \pm .03$ | $.73 \pm .04$ | $.15 \pm .04$ | $.58 \pm .04$ |
| FL | $.92 \pm .02$ | $.68 \pm .02$ | $\mathbf{.11 \pm .02}$ | $\mathbf{.42 \pm .02}$ | $.76 \pm .04$ | $.72 \pm .05$ | $\mathbf{.12 \pm .02}$ | $\mathbf{.45 \pm .05}$ |
| RS | $.91 \pm .03$ | $.68 \pm .03$ | $.12 \pm .03$ | $.46 \pm .05$ | $.75 \pm .05$ | $.73 \pm .05$ | $.14 \pm .04$ | $.53 \pm .06$ |

to more significant image understanding problems. Here, the verb classification models perform well on the 80%-100% certainty SQUID-E images, but top-1 accuracy falls drastically when human certainty drops to 60%-80%. Furthermore, we can see that for the images with labels above 40%, top-10 accuracy is substantially higher than top-1 accuracy, but below this threshold the difference between top-10 accuracy and top-1 accuracy decreases. This indicates that the models are much less likely to extract relevant event features for images rated below 40%. It should also be noted that the ResNet model trained on standard data (RN+SD) in Section 5.1 achieves higher accuracy scores than these situation recognition models' top-1 verb performance. This can likely be attributed to the fact that RN+SD is trained on four events while these models are trained on 504 verb classes.

## 5.3 Evaluating Uncertainty Quantification Methods

**Models**   We evaluate a collection of uncertainty quantification approaches on SQUID-E human judgments. We consider a selection of approaches that aim to reduce model overconfidence (label smoothing [45], belief matching [31], focal loss [44], and relaxed softmax [48]) as well as Monte-Carlo Dropout and a standard softmax + cross entropy loss baseline. We use the ResNet-based architecture described in Section 5.1 for the models in this experiment. We train two copies of each model, one using SQUID-E and one using the "high certainty" dataset introduced in Section 5.1.

**Task** We consider the task of binary classification where a model must identify whether or not an image belongs to a target event to mirror the annotation task detailed in Section 3.2. We compare mean human judgment scores of the SQUID-E validation set against the softmax logits of the trained models using mean squared error loss. We also report model accuracy on SQUID-E and the standard dataset (see Section 5.1), as well as the expected calibration error (using the ground truth labels). We run this experiment across 8 seeds and report the mean scores and standard deviation in Table 3.

**Results**   The model calibration techniques result in human judgment MSE improvements for the models trained on standard data, and more modest improvements for the models trained on SQUID-E. This is noteworthy considering that the models trained on SQUID-E achieve higher accuracy on the SQUID-E validation set. We hypothesize that this poorer calibration on RN+ES could possibly occur because RN+ES overall is less confident due to being trained on ambiguous data, whereas the calibration methods could have a larger impact on the more-confident RN+SD model. It should also be noted that the accuracy scores are slightly different from those in Section 5.2 because this is a binary classification task (to accurately compare against the human annotations which were collected through a binary decision task) while the task used in Section 5.2 was a 4-way classification task.

These results achieved using the models trained on standard data suggest that model calibration techniques can produce models that align better with human judgments. Similarly, they show that human uncertainty judgments can be used to directly evaluate calibration approaches. The MSE and ECE show positive correlation, indicating that comparing against human judgments aligns with more traditional calibration assessment metrics. Based on these results, work remains to be done to identify the best approaches for aligning model confidence with human judgments for noisy data.

# 6   Limitations, Ethical Considerations, and Alternate Approaches

**Limitations**   SQUID-E only includes human judgments for a small portion of its images, and so models cannot be directly trained on these judgments given the high variance within this domain. Furthermore, each annotated image only has 6 human uncertainty judgments, which is not enough samples to capture the distribution of human judgments for a given image. Experiments in Section 5 are consequently run on a small amount of data and may produce different results if run on different event types, etc. We also cannot guarantee that the frame selection algorithm detailed in Section 3.1 produced the optimal set of visually distinct frames. While we attempted to remove all videos that did not involve a particularly wide range of visual data or contained clips used in other videos, it is likely that not all were filtered out, and so there may be some overly-similar images in the dataset.

**Ethical considerations**   Our video collection queries were made in only 11 languages spoken widely online, and the majority were made in English, which likely produced an uneven distribution of regional and cultural representation within the dataset. Only collecting data via the YouTube platform also skewed the dataset's coverage. Because of these aspects of our data collection process, our dataset does not proportionately represent the experiences of the global population, which can potentially lead to biased models in downstream tasks. Similarly, we solicited our annotations from people located in the U.S. and used three-way redundancy for both tasks, meaning that the human judgments in the dataset are not representative of the general population, are likely skewed toward Western perspectives, and likely include demographic-driven biases.

While the owners of the videos from which we extracted images were not asked for permission to include their content in the dataset, all used videos have been made publicly available by their creators. If a creator takes their content offline the associated images will automatically be removed from our public dataset loader as well. It should be noted that the videos used to create the dataset were not checked for personally identifiable or offensive content.

**Alternate approaches**   There are multiple interpretations of what makes an image ambiguous, and what sort of ambiguous images are most relevant in the context of computer vision. In this paper, our goal is to minimize reporting bias [27] in image collection to retrieve a distribution of images that closely models natural human visual input. However, focusing on subsets of this distribution may provide data better suited for research in certain applications. For instance, some applications may only require images that depict the event type sufficiently well, and so images with low certainty scores should be removed from the main dataset. For other applications, it might be optimal to additionally remove images with high certainty labels and images with higher variance in their annotations. The remaining dataset, consisting of the images rated near the 50% mark with high agreement from annotators, could be used for efforts such as an in-depth analysis of the decision boundary of a model. These are both exciting directions for future work that consider different aspects of ambiguity and would provide opportunities for interesting analysis on these subsets of the dataset.

# 7   Conclusion

In this paper, we introduce a framework for generating datasets of ambiguous images and human uncertainty judgments and use it to develop SQUID-E (the Scenes with Quantitative Uncertainty Information Dataset for Events) consisting of 12,000 event-based ambiguous images and 10,800 human uncertainty judgments. We explore the characteristics of human uncertainty judgments for ambiguous visual data and show how this dataset can be used to assess the behavior of situation recognition models. Experiment results suggest that there is room for improvement in designing models that produce meaningful outputs when presented with ambiguous data, and that ambiguous datasets with human judgments can be used to train more robust models and to directly evaluate model calibration techniques. These findings motivate the creation of larger-scale ambiguous image datasets to develop more robust models and uncertainty quantification approaches and to further explore the relationship between models and human uncertainty judgments. This work also prompts other future work such as training models to learn individual annotators' uncertainty scoring functions and developing human-centric model calibration methods using human uncertainty judgments.

## Acknowledgments

The authors would like to thank David Etter, Elias Stengel-Eskin, Zhuowan Li, Zhengping Jiang, João Sedoc, and the anonymous reviewers.

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
