# A Dataset Construction Details

## A.1 Image Collection

The full list of events included in SQUID-E is: baseball, basketball, birthday parties, cooking, COVID tests, cricket, natural disaster fires, fishing, gardening, graduation ceremonies, hiking, hurricanes, medical procedures, music concerts, parades, protests, soccer/football, tennis, tsunamis, and weddings. Human uncertainty judgments were collected for the birthday party, wedding, parade, protest, COVID test, and other medical procedure event types. YouTube video queries were primarily made in English, but videos retrieved using Korean, Russian, Arabic, Chinese (simplified), French, Japanese, Hindi, German, Persian, and Spanish queries were also included.

36 frames from each video were sampled at even intervals, and each of these video frames were passed through a ResNet50 model [29] trained on ImageNet [59] attached to two pooling layers to featurize the frame. The feature vector of each sampled frame was passed into a k-means clustering algorithm with 6 centroids. The six frames whose featurizations had the closest Euclidean distance to a centroid were extracted and included in the dataset.

## A.2 Annotations

1,800 images were annotated using three-way redundancy on each task, resulting in a total of 10,800 uncertainty judgments (6 judgments per image). $0.20 was paid for six judgments (approximately $16/hr based on preliminary task completion time calculations), plus the 20% Mechanical Turk fee. This resulted in a total cost of $432 for the full set of human judgments, plus $31.50 for the initial pilot tests to identify quality annotators. Annotators were paid twice this amount for the intra-annotator variance analysis, resulting in a cost of $0.40 \times 10 \times 5 \times 1.2 = $24$. In all tasks, annotators were not given speed-related instructions or a time limit (other than the AMT task expiration limit). A screenshot of the annotation interface is included in Figure 5.

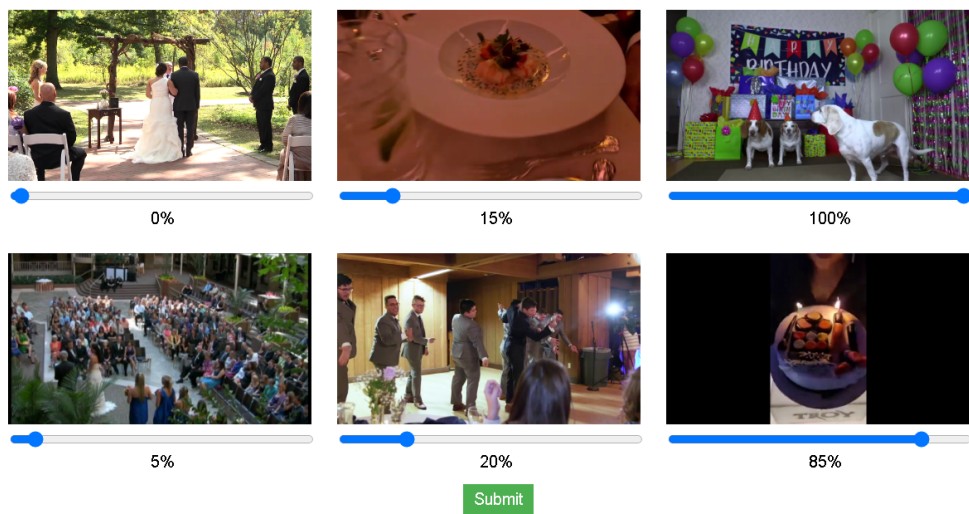

Figure 5: A screenshot of the human uncertainty judgment annotation setup. Annotators were provided with a target event type at the top of the page and were asked to use the sliding bars to rate their certainty that each of the six provided images belong to videos depicting that target event.

# B   Annotator Instructions

Below are the full instructions provided to annotators for human uncertainty labeling.

INSTRUCTIONS:

You will be presented with (1) a set of still images taken from videos and (2) a prompt specifying a type of event. Your task is to **rate your confidence that each still image belongs to a video depicting the provided event type** on a scale from **0% to 100%**. Rate each image individually. A guideline for ratings is shown in Table 4.

Table 4: Rating guidelines for annotators.

| Rating | Guidelines |
|---|---|
| **0%** | An image should only be rated **0%** if you are nearly certain that the video it belongs to does not depict the target event type. This rating would be appropriate if the image contains a set of attributes that, together, necessarily could not appear in the target event type. |
| **1% - 49%** | Rating an image between **1% - 49%** indicates that the visual evidence in the image suggesting it belongs to the target event type is weak enough that it is **likelier that the video depicts to another event type**. How weak the evidence is will determine where in the scale of 1-49 you rate it (keeping in mind the definitions of a **0%** rating and a **50%** rating). |
| **50%** | Rating an image at **50%** indicates that you feel there is an equal likelihood that the image belongs to a video of the target event type and that the image belongs to a video of a similar event type that shares some visual attributes (i.e., a birthday party and a wedding). |
| **51% - 99%** | Ratings between **51% - 99%** indicate that it is **likelier that the video depicts the target event than it doesn't**, and where on the scale you rate it depends on the strength of the visual evidence (again, keeping in mind the definitions of a **50%** rating and a **100%** rating). |
| **100%** | An image should only be rated **100%** if you are nearly certain that the video it belongs to depicts the target event type. This rating would be appropriate if the image contains a set of attributes that, together, could not reasonably belong to any other event other than the target event type. |

The event types you will be asked to consider in this task are **birthday parties**, **COVID tests**, **medical procedures** (other than COVID tests), **parades**, **protests**, and **weddings** (both ceremony & reception). Examples of core attributes belonging to these event types and images rated 100% for these event types are listed in Table 5. In addition to the attributes listed, you are encouraged to also draw from your own experiences when making confidence ratings.

Example ratings are shown in Table 6.

The event type is listed at the top of each page. Move the slider below each image to rate it. You may click on any image to increase its size to view image details more clearly. While the scores you assign are subjective in nature, we will be carefully checking to ensure that they follow the guidelines in the instructions. Please reach us at <email> if anything else is unclear or if you found an error in the task.

Table 5: Example images of events for annotators.

| Event Type | Images w/ 100% Rating |
|---|---|
| **Birthday party** |  |
| **COVID test** |  |
| **Medical procedure** |  |
| **Parade** |  |
| **Protest** |  |
| **Wedding** |  |

## C   Annotation Analysis

Historically, Spearman correlation has often been used for measuring agreement for scalar annotations [18, 61, 69]. However, other metrics, such as Fleiss's kappa and Krippendorff's alpha, are also methods of quantifying annotator agreement. Here, we compare these two metrics against Spearman correlation when applied to the annotations in SQUID-E. Fleiss's kappa requires nominal data, and so when computing this metric we bin the probabilistic judgments into categories (e.g., for 5 bins, annotations are divided into 5 classes: 0-20%, 20-40%, 40-60%, 60-80%, and 80-100%) and apply the metric accordingly. For Krippendorff's alpha, we use the interval metric for calculations. Results are reported in Table 7. The Spearman correlation and Krippendorff's alpha metrics align closely, whereas Fleiss's kappa is consistently much lower. We hypothesize that this is the case due to binning being an imperfect method of converting quantitative data to nominal data.

Table 6: Rating examples for annotators.

| RATING EXAMPLES |
| --- |

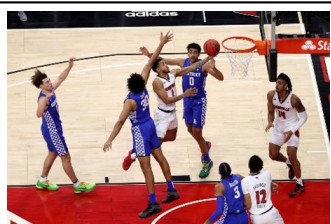

**WEDDING**
**Rating: 0%.** All visual attributes in this image suggest that the video is of a basketball game, which virtually never coincides with a wedding event in the same video.

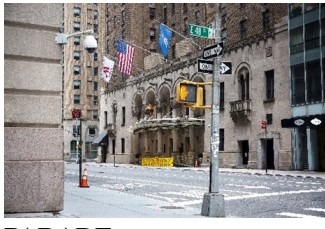

**PARADE**
**Rating: 10%.** Depicts the location of a parade, but lacks all distinguishing attributes of a parade. Could conceivably belong to a video of this event type, but there are many possible events that are significantly more likely.

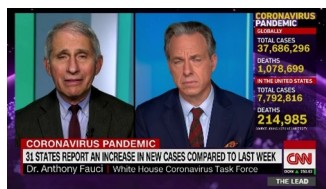

**COVID TEST**
**Rating: 30%.** Contains attributes closely related to a COVID test, but is not an image that would occur immediately before/after a depiction of a COVID test in a video, making it less likely than an image that would score 50%+.

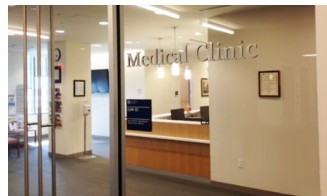

**MEDICAL PROCEDURE**
**Rating: 50%.** The image could quite possibly belong to a video depicting a medical procedure, but there are few enough defining features that it could easily belong to a different event type as well (i.e. a video tour of the clinic, a news story about hospital staff shortages, etc).

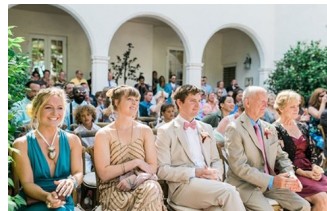

**WEDDING**
**Rating: 85%.** Has many attributes closely tied to a wedding, but could conceivably belong to a closely related event type.

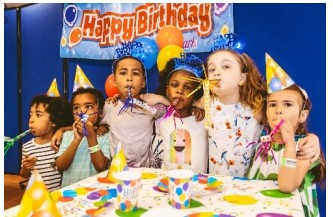

**BIRTHDAY PARTY**
**Rating: 100%.** Most of the attributes in this image are uniquely characteristic of a birthday party. The chances of these elements occurring together in another setting are virtually nonexistent.

# D   Experiment Details

All experiments are run on an internal cluster using 1 GPU and 12 GB of memory. Experiments described in Sections 5.1 and 5.2 are run using annotations from task variant A, and experiments described in Section 5.3 are run using annotations from task variant B to allow for both positive and negative samples to be used in evaluation.

## D.1   Training

**Section 5.1**   We use a headless ResNet50 model attached to a fully connected layer for all three models. They are initialized with ImageNet weights, as weights pretrained on the SWiG situation recognition dataset [52] resulted in poorer overall performance. Training and validation sets are mutually exclusive for both datasets, and the SQUID-E validation set does not include frames from videos that are included in the SQUID-E training data. For this experiment we evaluate models on both datasets. We train each model for 5 epochs with a learning rate of 1e-5 using the Adam

Table 7: Agreement scores for the human annotations in SQUID-E across the two task variants using various agreement metrics. Spearman is considered in Section 4 of the paper, Alpha refers to Krippendorff's alpha, and Kappa refers to Fleiss's kappa. When computing Fleiss's kappa, we converted the quantitative scores to nominal data by binning. We consider this metric when using 3, 4, and 5 bins.

| Task | Spearman | Alpha | Kappa (3 bins) | Kappa (4 bins) | Kappa (5 bins) |
|------|----------|-------|----------------|----------------|----------------|
| A | .676 | .658 | .468 | .397 | .341 |
| B | .631 | .696 | .491 | .431 | .386 |
| A+B | .673 | .676 | .482 | .424 | .364 |

optimization algorithm. Below, we describe the unique TorchVision augmentation filters applied to each model:

RN+SD: No data augmentation.

RN+PA: `ColorJitter(0.5,0.5,0.5,0.5)`, `RandomSolarize(220)`, `RandomPosterize(4)`.

RN+GA: `Resize(512)`, `RandomPerspective(0.5)`, `RandomCrop(256)`.

RN+NM: `RandomErasing(0.5)` (applied twice), `GaussianBlur(kernel_size=(5,9))`.

RN+AU: `RandomPerspective(0.5)`, `RandomCrop(256)`, `RandomErasing(0.5)`, `GaussianBlur(kernel_size=(5,9))`.

RN+AM: `AugMix(5)`.

**Section 5.3** The models are trained on a dataset of 960 event-centric images, where 480 belong to the target class, and the other 480 are equally comprised of three other event types. The validation datasets both consist of 120 images of the target event and 120 images of a similar but distinct event (e.g. "birthday party" and "wedding" or "parade" and "protest"). We train each model for 5 epochs with a learning rate of 1e-5 using the Adam optimization algorithm.

## D.2    Section 5.2 Verb Prediction

In the situation recognition task, an event is defined by (1) a verb (e.g. jumping) and (2) the set of semantic roles dependent on that verb (e.g. agent: boy, source: rock, destination: water) [78]. As stated in Section 2, contemporary models first predict the event's verb and then pass that verb into a semantic role classification model to predict the event's roles. Therefore, a situation model's accuracy is upper-bounded by its verb prediction accuracy. Given this, we simplify the task of situation recognition in this experiment by focusing solely on models' verb classification performance. We take the verbs assigned to each event using the ImSitu event ontology (used to train most contemporary situation recognition models) and assess performance by identifying how accurately models can predict these verbs.

## D.3    Alternate Binning Approach

While we primarily consider mean human annotation scores for the experiments in Section 5, some literature argues for handling human quantitative judgments differently. Peterson et al. consider the labels of a data point as samples from an underlying label distribution [51], whereas Basile et al. propose that, for subjective tasks, all annotations may be "correct" and should therefore all be used in evaluation without pre-aggregation [7, 8]. They propose an evaluation method where model outputs are compared against each annotation label individually. Along these lines, in this experiment we consider each (image, judgment label) pair as a data point for binning and compare the resulting accuracy scores against what is currently listed in Table 2. Results of this experiment are reported in Table 8. As shown in the table, the alternate binning method increases model performance for all bins but 80-100%, which drops in accuracy, but overall the same performance trends remain.

Table 8: Accuracy of situation recognition models on SQUID-E extending the experiment described in Section 5.2 and reported in Table 2. Rows with "mean" bins reflect the original experiment setup, and rows with "alt" bins treat every (image, annotation) pair as its own data point when binning. Accuracy of the top scoring verb as well as the top 10 scoring verbs are reported (listed as "Top 1" and "Top 10" respectively). Best results for average accuracy are listed in bold.

| Model | Bins | 0-20% | 20-40% | 40-60% | 60-80% | 80-100% | Avg. |
|---|---|---|---|---|---|---|---|
| *Verb Accuracy (Top 1)* | | | | | | | |
| JSL | Mean | .00 | .07 | .17 | .22 | .52 | .35 |
| GSRTR | Mean | **.02** | .09 | **.22** | **.25** | **.59** | **.41** |
| CoFormer | Mean | **.02** | **.13** | **.22** | .23 | .58 | .40 |
| JSL | Alt | .04 | .09 | .19 | .32 | .49 | .33 |
| GSRTR | Alt | **.07** | .11 | .19 | .39 | **.55** | **.38** |
| CoFormer | Alt | **.07** | **.15** | **.20** | **.40** | .54 | **.38** |
| *Verb Accuracy (Top 10)* | | | | | | | |
| JSL | Mean | **.11** | .43 | .49 | .72 | .86 | .66 |
| GSRTR | Mean | **.11** | **.55** | .54 | .77 | .88 | **.70** |
| CoFormer | Mean | .09 | .42 | **.58** | **.82** | **.91** | **.70** |
| JSL | Alt | .23 | .49 | .61 | .73 | .83 | .66 |
| GSRTR | Alt | **.26** | **.54** | **.72** | .77 | .85 | **.70** |
| CoFormer | Alt | .23 | .50 | .67 | **.82** | **.88** | **.70** |

Table 9: Results showing a comparison between taking the MSE and taking the KL divergence of the calibrated model logits and the human certainty scores. ECE is also provided for additional context. As shown below, the results suggest a positive correlation between the three metrics.

| | Trained on SD | | | Trained on SQUID-E | | |
|---|---|---|---|---|---|---|
| | HUJ MSE | KL | ECE | HUJ MSE | KL | ECE |
| Baseline | .15 ± .05 | .49 ± .17 | .58 ± .02 | .16 ± .04 | .52 ± .13 | .61 ± .05 |
| Monte Carlo | .14 ± .05 | .45 ± .16 | .57 ± .03 | .16 ± .04 | .46 ± .11 | .57 ± .04 |
| Label Smoothing | .12 ± .03 | .30 ± .08 | .46 ± .02 | .14 ± .03 | .36 ± .08 | .52 ± .04 |
| Belief Matching | .14 ± .05 | .42 ± .14 | .55 ± .02 | .15 ± .04 | .45 ± .11 | .58 ± .04 |
| Focal Loss | **.11 ± .02** | **.29 ± .06** | **.42 ± .02** | **.12 ± .02** | **.31 ± .05** | **.45 ± .05** |
| Relaxed Softmax | .12 ± .03 | .32 ± .08 | .46 ± .05 | .14 ± .04 | .37 ± .09 | .53 ± .06 |

## D.4 Mean Squared Error vs. KL Divergence

For the experiment detailed in Section 5.3, we additionally calculated the KL divergence between the model confidence scores and the human judgments to compare against the MSE. Results are aggregated across 8 seeds and are presented in Table 9. As shown in the table, there is a positive correlation between the three metrics, indicating that the MSE and KL divergence measure similar aspects of the model's calibration.

## E  Asset Licenses

**Models**    JSL [52] is licensed under the MIT License, and GSRTR [17] and CoFormer [16] are licensed under Apache License 2.0.

**Datasets**    Visual Genome [35] is licensed under CC-BY 4.0, WIDER [77] is available for research purposes, and UCLA Protest Images [75] is available for academic use only.