# OpenReview forum: "Ambiguous Images With Human Judgments for Robust Visual Event Classification"
_NeurIPS.cc/2022/Track/Datasets_and_Benchmarks — NeurIPS 2022 Datasets and Benchmarks _

### Official Review · Reviewer_gzHA · 2022-07-19
**Hard to follow text, not convincing image collection procedure**

**Rating:** 4
**Confidence:** 3

**Strengths:**

The authors tackle a relevant problem in the literature and propose a setup where humans can annotate a "human judgement score" to represent their confidence of an event type based on a snapshot from a video sequence.

**Weaknesses:**

It is not clear whether the image collection procedure is ideal. Authors state in line 551 that if number of time frames for a given video permit, every 240th frame was sampled. If not, then a clustering algorithm is run on 36 evenly sampled frames from a video to decide on the 6 frames to be used from the given event type video. If the former case applies, the snapshots in the datasets can be very arbitrary. If the latter case applies, (i) this implies the video is less than 48s for a 30fps vid, and (ii) the sampled 36 frames are simply ~1s apart snapshots that can again contain arbitrary content, not necessarily strictly related to the event type that the snapshots will be labeled as.

It is not clear whether the annotation procedure is optimal for best value and there are ambiguities. Line 175, variant A implies that an annotator at a given time, gives human judgement score for 6 images of the same event type, which count towards 6 separate annotations. Variant B, however, asks annotations for a given event type despite showing 3 images of another event type. Are the annotations for the other event type ignored? How are they aggregated into the annotations? Based on the manuscript, it seems that each test sample has six annotations of judgement score only for the corresponding class.

A major missing information on the collected data is regarding the standard deviation of uncertainty judgements for images which have a low median judgement score. This could give insight as to whether low judgement scoring samples are in agreement across the 6 annotators or not. Low variation among low judgement scoring samples could imply the sample is simply unrelated to the event type.

It is not clear how the bins in experiments are created: For example, when considering samples that had 0-20% human-judged ambiguity, do the authors look at the average judgement score across 6 annotators and see if it falls within interval [0,20]? If so, is the mean value the right metric to aggregate?

Sec 5.2. experimental setup is not clear. It seems that different compared setups (SD vs AUG vs DAI) use different number of sample and classes to train the models. From paragraph starting at line 253, it would seem that RN+DAI would be trained on 7 event classes whereas others would be trained on several hundred classes. However, the paragraph starting on line 261 state that RN+SD is only trained on four event types. Furthermore, line 266 mentions "top-5 scores", which would not make sense if the models are trained on four event types. Is the top-5 of Table 1 is compared with top-1 of Table 2?

It is not clear what is the advantage of training on DAI if the training set has no HUJ. If the authors believe training models with samples that are loosely related to an event type is beneficial, they should potentially benchmark their work with respect to a collection of different preprocessing/data augmentation schemes.

**Additional Feedback:**

Potential typo: Fig. 4 caption, "high uncertainty", or received a mean human judgement score at or above 95%,...

**Clarity:**

Hard to follow overall, especially the Experiments section. The discussion over the results need to be rephrased and the points authors want to make should be clarified.


**Correctness:**

There seems to be a crucial flaw with both the collection and the annotation of the datasets.
Image collection is done in an automatic way using unconvincing heuristics, with a high chance of collecting unrelated snapshots from a given event type video.
If Variant B is used during annotation, then it implies there are either effectively half the number of annotations for each event type, or authors apply a heuristic not mentioned in the manuscript to account for the lost annotations. Furthermore, it is not clear what percent of the test set is annotated using Variant A (or B).


**Documentation:**

Line 258: Used augmentation filters should be clarified in the appendix.


**Ethics:**

There are no relevant ethical concerns. The fact that certain languages and hence cultures could be ignored when creating this dataset should have limited impact due to dataset size being already rather small to be used for training purposes itself.


**Relation To Prior Work:**

Authors list some key literature from recent years where aleatoric and epistemic uncertainties are modeled through different preprocessing of data and optimization procedures. Furthermore, authors also separately refer to works in cognitive science regarding human uncertainty judgements and state that works in similar nature to this manuscript are very rare.

Nevertheless, novel contributions in this manuscript are not clearly distinguished with respect to findings of prior works.

**Summary And Contributions:**

This work presents a method to generate dataset of ambiguous images (DAI) which focuses on images with challenging content. Accordingly, authors generate 12k ambiguous samples, out of which a test set of 1.8k samples have 6 manual annotations of human uncertainty judgements each. The authors empirically show that using DAI improves results for ambiguous samples and the human uncertainty can be used for model calibration.

---

> ### Author Response · Authors · 2022-08-18
> **Thank you for your review. Regarding data collection, novel contributions, and other clarifications (1/2):**
>
> Thank you for the detailed review. We have addressed a few common concerns in the general comment, and have answered your personal comments and feedback below. We are currently updating the paper to incorporate feedback and this will be uploaded before the revision deadline.
>
> > When long videos are sampled, no clustering algorithm is applied and the extracted frames can be very arbitrary.
>
> It was not sufficiently clear in the original submission, but we applied the ResNet-based clustering algorithm to each video to produce the set of six frames. This will be clarified in the appendix.
>
> > The reviewer states that the image collection procedure is not ideal, as it contains images that may not be explicitly relevant to the ground truth event type.
>
> In this paper we believe that highly ambiguous images that do not clearly depict the event type are important data points to include in our dataset. Our aim is to explore the full range of visual input associated with a given event type, even if not all of the images directly depict important aspects of that event. Therefore, we are able to accomplish this by sampling frames from each video and clustering them to extract the most visually diverse set of images while also mitigating dataset reporting bias as described in Section 3.1.
>
> > Which annotations (from Task A, Task B, or both) are used for the experiments?
>
> We use Task A annotations for binning in experiments in sections 5.1 and 5.2, and use Task B annotations for calculating the HUJ MSE in the experiments in section 5.3 to allow for positive and negative samples to be used for evaluation. This will be clarified in the appendix.
>
> > Do the annotations for images with low average judgments have high agreement? If so, this could imply that the image is not related to the ground truth event.
>
> When manually filtering videos as described in Section 3.1, we aimed to remove videos that depicted content not pertaining to the event type. Given this, as described in the first portion of Section 3.1 we believe that the low-certainty images are important to include even if they do not directly depict the key elements of the event type in order to produce a low-bias representation of visual data related to the event type. The images with a lower mean score (<20) have a mean standard deviation of 6.06 while the mean standard deviation across the dataset is 14.31, suggesting that there is more agreement for these images (partially enforced by the mean being <20). Analyzing these low-certainty images further would provide interesting insight, but in this paper we leave this as future work.
>
> > Is the mean annotation score used to aggregate annotations for binning purposes? If so, is the mean an appropriate metric to use?
>
> Yes, we aggregate annotations using the mean to produce bins for the experiments in Sections 5.1 and 5.2. Researchers have proposed a few solutions for evaluating models with noisy labels. Peterson et al. consider the labels of a data point as samples from an underlying label distribution [1], whereas Basile et al. propose that, for subjective tasks, all annotations may be “correct” and should therefore all be used in evaluation without pre-aggregation [2, 3]. They propose an evaluation method where model outputs are compared against each annotation label individually. Along these lines, it would be interesting to consider each (image, judgment label) pair as a data point for binning in the Section 5.1 and 5.2 experiments and compare the resulting accuracy scores against what is currently listed in Section 5. We will add this analysis to the experiments section in the updated revision.
>
> > The experiment setup for Section 5.2 is not clear. How many event types are each model trained on, and how are the results from Section 5.1 and Section 5.2 compared?
>
> RN+SD, RN+AUG, and RN+DAI are all trained on 4 event types and the same number of images. The only difference is that the images used to train RN+SD and RN+AUG are sourced from existing image datasets while those used to train RN+DAI are from DAI. Additionally, the comparison on line 266 is between the top-5 scores in Table 1 to the top-1 scores of Table 2, but we have since removed this comparison from the paper to focus on more direct comparisons between RN+DAI and other data. We are rewriting the experiments section (Section 5) to improve clarity in experiment setup, motivation, and major takeaways.

---

> > ### Author Response · Authors · 2022-08-18
> > **Thank you for your review (2/2)**
> >
> > > What are the primary takeaways of Section 5.2? It would be informative to include a comparison between a model trained on DAI and models trained using various data augmentation techniques.
> >
> > Thank you for the feedback. The primary motivation of Section 5.2 is to show that training on ambiguous images can produce models that are more robust to other ambiguous data compared to other approaches. We agree that using a larger set of augmentation approaches would make the experiment in Section 5.2 more meaningful. We are running experiments that compare the model trained on DAI against a wider variety of augmentation approaches.
> >
> > > The paper’s novel contributions are not clearly stated in the related works section.
> >
> > Our primary novel contributions with respect to prior work are:
> >
> > (1) We present the first dataset of intentionally ambiguous event-centric images. We introduce a method to collect images that mitigates the reporting bias found in existing image datasets to model the distribution of visual input experienced by humans. We argue that having datasets that more closely model this distribution is important for developing models that are robust to the range of visual data present in many real life scenarios, such as human-robot collaboration, and to our knowledge this is the first dataset that attempts this.
> >
> > (2) Furthermore, to our knowledge this is the first event-centric image dataset that uses human uncertainty judgments as labels. We show in the experiments that these labels can be used to assess model accuracy on different distributions of ambiguous data and to directly assess model calibration techniques.
> >
> > We are revising the related work section to emphasize these contributions and how they differ from prior work in the updated revision.
> >
> > > Line 258: Used augmentation filters should be clarified in the appendix.
> >
> > The augmentation filters are detailed in lines 603-605 in the original submission, but this section will be expanded on once we have finished running additional augmentation experiments.
> >
> >
> > [1] Peterson, Joshua C., Ruairidh M. Battleday, Thomas L. Griffiths, and Olga Russakovsky. "Human uncertainty makes classification more robust." In Proceedings of the IEEE/CVF International Conference on Computer Vision, pp. 9617-9626. 2019.
> >
> > [2] Basile, Valerio, Michael Fell, Tommaso Fornaciari, Dirk Hovy, Silviu Paun, Barbara Plank, Massimo Poesio, and Alexandra Uma. "We Need to consider disagreement in evaluation." In 1st Workshop on Benchmarking: Past, Present and Future, pp. 15-21. Association for Computational Linguistics, 2021.
> >
> > [3] Basile, Valerio. "It’s the end of the gold standard as we know it. on the impact of pre-aggregation on the evaluation of highly subjective tasks." In 2020 AIxIA Discussion Papers Workshop, AIxIA 2020 DP, vol. 2776, pp. 31-40. CEUR-WS, 2020.

---

> > > ### Comment · Reviewer_gzHA · 2022-08-24
> > > **Thank you for your clarifications**
> > >
> > > Thank you for your responses. I look forward to the revised manuscript. I have a few standing questions/concerns:
> > >
> > > >In this paper we believe that highly ambiguous images that do not clearly depict the event type are important data points to include in our dataset. Our aim is to explore the full range of visual input associated with a given event type, even if not all of the images directly depict important aspects of that event. Therefore, we are able to accomplish this by sampling frames from each video and clustering them to extract the most visually diverse set of images while also mitigating dataset reporting bias as described in Section 3.1.
> > >
> > > I am not sure if this is a good idea especially when the sampled frames are not human verified. For example, the video can have some unrelated advertisement, or some other style that incorporates unrelated scenes due to the creators desire or other factors. This outcome does not even have to be malicious and can be present in any publicly available video. What is concerning is that without human in the loop (not against false positive videos, but for each selected frame), such scenes will most likely be picked up by the clustering algorithm you mention.
> > > In a way, I am afraid the finding that there is significantly lower standard deviation (6.1 as opposed to 14.3) in images with lower judgement score (<20) corroborates my concern with the image collection procedure.
> > >
> > >
> > > In addition, I would appreciate you can also clarify the procedure of creating the annotations from human scores for both Variant A and B.

---

> > > > ### Author Response · Authors · 2022-08-25
> > > > **Revision uploaded**
> > > >
> > > > Thank you for your response. We have uploaded the revised manuscript. To respond to your comments:
> > > >
> > > > > The video can have some unrelated advertisement, or some other style that incorporates unrelated scenes.
> > > >
> > > > When collecting the videos for this dataset, we aimed to remove videos that included cuts to completely unrelated clips such as advertisements. Therefore, these low-certainty, high-agreement images, while not particularly representative of a given event, are in general likely to be part of the distribution of visual data that a human would observe when watching a video of that event type. Adding additional human filtering of individual images would introduce human reporting bias that we discuss with respect to image datasets in Section 3. By including all images selected by the algorithm without a human-in-the-loop verification system, we avoid bias and produce a dataset that better reflects the true distribution of visual data pertaining to videos of these events.
> > > >
> > > > > Such scenes will most likely be picked up by the clustering algorithm.
> > > >
> > > > As shown in Section 4, low-scoring images make up a modest portion of the dataset. Images that receive a mean confidence score of 20% or below in task A make up 21.8% of the annotated images, which is approximately proportional.
> > > >
> > > > > In addition, I would appreciate you can also clarify the procedure of creating the annotations from human scores for both Variant A and B.
> > > >
> > > > For human annotations, in the dataset we provide individual raw human judgment scores labeled with their corresponding image, task variant, event prompt, and image ground truth label. For the experiments in Section 5.1 and 5.2, we take the mean of the three human judgments collected for each image in task variant A, and use these resulting values for binning in Tables 1 and 2.
> > > >
> > > > For comparing model logits to human uncertainty scores in Section 5.3, we consider a binary classification task where the model must determine whether or not an image depicts event type X. Each model is validated on an image set that consists of 50% positive image samples, which depict event X, and 50% negative image samples, which do not depict event X. We choose this distribution of validation data because only validating with positive data samples may not produce a representative assessment of a model’s uncertainty quantification.
> > > >
> > > > For each of the images in this set, in task variant B, annotators were asked to give their confidence that the image (regardless of its ground truth label) belonged to event X. For each image, we take the mean human score produced in this task and assign that as the human label. These labels are used as the human annotations for this experiment that model logits are compared to.

---

> > > > > ### Comment · Reviewer_gzHA · 2022-08-26
> > > > > **Thank you for your response.**
> > > > >
> > > > > >Such scenes will most likely be picked up by the clustering algorithm.
> > > > >
> > > > > > > As shown in Section 4, low-scoring images make up a modest portion of the dataset. Images that receive a mean confidence score of 20% or below in task A make up 21.8% of the annotated images, which is approximately proportional.
> > > > >
> > > > > Am I misinterpreting your answer, or are you agreeing that scenes with mean confidence score 22% or below are likely to be scenes unrelated to the event type?

---

> > > > > > ### Author Response · Authors · 2022-08-26
> > > > > > **Clarification**
> > > > > >
> > > > > > Thank you for raising this question. No, by presenting this statistic we intended to clarify that the clustering algorithm doesn’t disproportionately select low-scoring frames to add to the dataset. As stated in our previous response, we believe that given our video filtering method the majority of these low-scoring frames do belong to scenes related to the event type and belong to the distribution of visual data we hope to model.

---

### Official Review · Reviewer_eRWV · 2022-07-21
**Ambiguous images classification**

**Rating:** 7
**Confidence:** 4
**Clarity:** Yes

**Strengths:**

- The topic is of high importance and often not considered
- The work is clearly structured and gives a good motivation
- The method and results are described in detailed and thus are easy to follow
- they investigated different options like the task variants A & B
- The collection process is easy and reprocducible. I especially liked the criteria for selecting appropriate annotators because it was easy and efficient.
- Issues like the lower performance of RN+SD in line 263 are highlighted and appropriately discussed.

**Weaknesses:**

- The evaluation is mainly based on the metric Accuracy (ACC). However the authors want to measure the difference between two distribution. In my opion a score like the Kullback-Leibler Divergence would be more suited. Additionally, the expected calibration error (ECE) (https://arxiv.org/pdf/1706.04599.pdf) would also  be very suited for the analysis.
- The authors discuss the issue of overconfident models and I would recommend adding references to ECE (a metric to measure this overconfidences) at these parts
- The standard deviations are missing which limits the interpretability. This makes it for example difficult to estimate the impact of the uncertainty quantification methods in Table 3.
- As discussed in the limitation only six annotations per image were used and the uncertainty assingments show an interperson variablity. These issue limit the generalizability of the scores.
- Additionally, the issue of intraperson variablitly (over time) is not discussed. I would expect that the same person will rate the same image differently after a while. This issue impacts the complete collection and as such must be addressed. Especially,  in the source of annotator disagreement this topic was not mentioned.
- The authors defined and annotated 20 event types. However often only a selection of classes is evaluated. For example in setion 5.2, why do you use 4 classes instead of all 20. If there is a reason please state it otherwise it looks like you selected the 4 best working classes for demonstration purposes (cherry picking).


**Additional Feedback:**

Please verify that you have the right to use the selected videos to create a new datasets. I'm uncertain if intelluctual property or any other copyright you prevent from the legal usage.

**Correctness:**

- In line 191 you state that inter-annotator variance is a form of epistemic (model-driven) uncertainty. I disagree on this conclusion. Inter-annotator variance should be considered aleatoric uncertainty (contained in the data). The epistemtic uncertainty could be lowered with more training samples, however in your example the poor label quality due to the missing uncertainty quantification limits the model performance. The discussion about inter- and intraoberserver variance is also important here because this illustrates that the uncertainty is already present in the data.

**Documentation:**

- The dataset is described but for ease of use not already containted in the supplementary material.
- The installation requirements in the experiments is limited with not even 50 words which makes it more difficult to reproduce

**Ethics:**

Appropriate concerns regarding the datasources are discussed.

**Relation To Prior Work:**

In general the background is adequatly described.

However, some in my opinion highly related papers are not discussed at e.g. other ambiguous datasets (CIFAR10H https://arxiv.org/abs/1908.07086), discussion about ambiguity noise in current dataset (http://arxiv.org/abs/2006.07159), alternative treatments of ambiguity via clustering (https://www.mdpi.com/1424-8220/21/19/6661) and benchmarks for uncertainty quantification (https://arxiv.org/abs/2106.04015).

**Summary And Contributions:**

The authors discuss the issue of uncertainty inherent in human annotations. In most cases the uncertainty is not quantified and thus ignored during the training. This leads to a poor model performance due to noisy images and the underyling aleotric uncertainty.

The authors propose a new collection process for ambiguous images with uncertainty quantification, present at dataset of 12,000 images collected via the proposed process and quantify the impact of uncertainty on the classificaiton.

---

> ### Author Response · Authors · 2022-08-18
> **Thank you for your review. Regarding related work, experiment design, and other clarifications:**
>
> Thank you for your thorough review and feedback. We have answered some common questions in the general comment, and respond to your specific concerns below. We are also currently updating the paper to incorporate feedback.
>
> > The reviewer provides a set of four papers that cover a range of work related to this paper.
>
> Thank you for these citations. We agree that they are highly relevant to our work and will be included in the related works section. We would like to address a few key differences between these works and our paper here. CIFAR-10H is a highly applicable image dataset that successfully leverages human judgment labels, but it differs from our work in several ways. Firstly, the images that comprise CIFAR-10H (1) depict single objects and (2) are not intentionally collected to be ambiguous. Much of the ambiguity that characterizes these images stems from the low resolution of the images, whereas the ambiguity of images in DAI stem from a wide range of sources due to the complexity and compositionality of the event-based images. Furthermore, Peterson et al., Beyer et al., and Schmarje et al. consider data labeled with sets of discrete classification labels, while in this paper we consider probability-based uncertainty quantification scores. Finally, in Nado et al.’s paper on uncertainty quantification benchmarking, the evaluation systems implemented rely on ECE and KL-divergence as opposed to direct comparison against human quantification.
>
> > The images in the dataset are not directly included in the supplementary materials, and it isn’t clear that we have the right to use these videos in our dataset.
>
> Yes, we do not own or have legal rights to the videos used for this dataset, and so we cannot distribute them directly. Our dataset explicitly consists of the annotations for these videos, namely the video URLs, frame numbers, ground truth values, and human uncertainty judgments. Therefore, we only distribute these annotations and a script to download the videos locally. More information regarding our usage of these videos is included in Section 6 of the paper.
>
> > Not every class is used for evaluation in Section 5. “If there is a reason please state it otherwise it looks like you selected the 4 best working classes for demonstration purposes (cherry picking).”
>
> We want to emphasize that we do not cherry-pick these event types - these are the only sets of event types that we can use to run the Section 5 experiments. We train the models in Section 5.2 and 5.3 on 4 event types because these are the only four events that both have human judgment scores and are sufficiently represented in high-certainty image datasets. Similarly, the two event types used for 5.1 are the only two that both have human annotation scores and can be defined using the event ontology used by the state-of-the-art models evaluated in that experiment. Given these restrictions, these are the only sets of events we run our experiments on.
>
> > Inter-annotator variance should be considered as aleatoric uncertainty.
>
> Thank you for pointing this out. We have removed this claim from Section 4 of the revised paper. It can be argued that each annotator can be viewed as an individual classification model with intrinsic biases driven by the factors outlined in section 4.2, and so the variance that stems from these biases would be epistemic. However, this variance is also largely influenced by the aleatoric uncertainty driven by the noisiness in the data, and so we agree that it is not true that this variance is purely a form of epistemic uncertainty. We will update the paper accordingly.
>
> > KL-divergence and ECE should be used as metrics in the experiment in Section 5.3.
>
> We agree that this would be an interesting addition to this experiment. We are re-running this experiment with additional metrics added and will report results in the paper revision.
>
> > Error bars should be added to the experiments.
>
> If time permits, we will run experiments across multiple seeds and add the error bars where applicable in the revision.
>
> > The intra-person variation of annotations in the dataset should be discussed, especially in Section 4.2.
>
> We agree that this is an important metric to consider when discussing inter-annotator variance and the annotations as a whole. We will include an analysis of this in the revision. We are also running a small study using Amazon Mechanical Turk annotators to characterize this variance and this will be added to the paper revision if a sufficient number of annotations can be collected in time.
>
> > “The installation requirements in the experiments is limited with not even 50 words which makes it more difficult to reproduce”
>
> We are updating the supplementary materials to make it easier to reproduce the experiments in the paper and will upload the revised zip file along with the paper revision.

---

> > ### Comment · Reviewer_eRWV · 2022-08-25
> > **Revision**
> >
> > Thank you for addressing my concerns. I will comment below on some of your answers however most parts I can not evaluate due to the fact that you have not uploaded a revision. Please upload a revision including the updated supplementary so that I may verify your claims and potentially edit my score.
> >
> > **The reviewer provides a set of four papers that cover a range of work related to this paper.** Do you plan to discuss the mentioned topics in revised version of your related work?
> >
> > **Yes, we do not own or have legal rights to the videos used for this dataset, and so we cannot distribute them directly.** I saw your clarification and think this is a possible solution. However, it limits the usability of the benchmark if videos would be removed in later version. Do you plan on keeping records of the currently usable datasets e.g. via versioning if anything happens?

---

> > > ### Author Response · Authors · 2022-08-25
> > > **Revision uploaded**
> > >
> > > Thank you for your response. We have uploaded the revised manuscript and supplementary material. To respond to your questions:
> > >
> > > > Do you plan to discuss the mentioned topics in revised version of your related work?
> > >
> > > Yes, we include a discussion surrounding these papers and their ideas in the related works section.
> > >
> > > > Do you plan on keeping records of the currently usable datasets e.g. via versioning if anything happens?
> > >
> > > Yes, we will be using GitHub to distribute the dataset which will allow for transparent versioning if we need to update the dataset.

---

> > > > ### Comment · Reviewer_eRWV · 2022-08-26
> > > > **Revised Paper**
> > > >
> > > > I thank the author for uploading the improved revised work. I believe all of my concerns were addressed. I will raise my score from 6 to 7 to reflect this change.

---

### Official Review · Reviewer_vrs4 · 2022-07-25
**Ambiguous Images With Human Judgments for Robust Visual Event Classification**

**Rating:** 7
**Confidence:** 3
**Correctness:** The results are reasonable and show n…
**Clarity:** The paper is well-written in general …

**Strengths:**

- The paper is well written and clearly understandable.
- The authors contribute a human uncertainty dataset of 12,000 images with human uncertainty judgements.
- The authors conduct extensive evaluations with different state-of-the-art methods to have a larger fundament for the evaluation.

**Weaknesses:**

The authors successfully show that a model trained on DAI is indeed more accurate on such data (while being less accurate on standard data). However, the experimental results do not strongly indicate that ambiguous training data leads to more robust results or approximates human uncertainty in general. There are numerous studies that the usage of soft labels in training can enhance the robustness in classification tasks. While I appreciate the extensive evaluation of the study, the derived take-home message has no strong new insight.

**Additional Feedback:**

A stronger emphasis on how to interpret the results and how they are significant would be helpful to strenghten the impact of your study. How do you interpret the smaller HUJ MSE, when the model is trained on standard data? The 'Experiments' section should start with a brief overview of the three experiments, why they are done and how they are connected to simplify the readability.

**Documentation:**

The dataset is not publicly available, but will be released on Github on acceptance. The datasheet contains the intended use case and a recommendation of how to use the data, licensing information and a maintenance plan.

**Ethics:**

The utilized data are publicly available and the contributed methods do not anyhow create ethical concerns.

**Relation To Prior Work:**

The authors position themselves suitably in prior work and present state-of-the-art methods in their application fields.

**Summary And Contributions:**

The authors propose a new method to generate ambiguous imagery of specific events with human judgements to create a dataset for a robust event recognition benchmark. By applying their strategy, they construct DAI, a dataset of ambiguous 12,000 images, 1,800 of these with human uncertainty judgements. The authors claim that event classification models trained on ambiguous data are more robust to ambiguous images. To demonstrate that, they apply the following steps:

1. The authors use different verb predictors (JSL, GSRTR, CoFormer) on DAI to generate verbs with their probability from ambiguous event images in DAI and transfer them to events using the FrameNet ontology.
2. The authors compare the generated event probabilities with their corresponding human uncertainty judgements and show expected correlations. They show that a higher number of verbs significantly supports the allocation to the true event type (5 vs 1).
3. The authors compare the impact of different training datasets and demonstrate that verb classifiers have the highest accuracy on DAI when they are trained on DAI (9% higher than augmented standard data). They also show that the mean standard error against human judgements is smaller when trained on standard data, which seems contradictory to their hypothesis that training on ambiguous data will  lead to results that resemble human uncertainty.

---

> ### Author Response · Authors · 2022-08-18
> **Thank you for your review. Regarding experiment results:**
>
> Thank you for your thorough review. We provide clarification regarding your comments below, and are revising the paper to address these concerns and other suggestions. We also address some common concerns from reviewers in the general comment.
>
> > What are the primary takeaways of the experiments in Section 5, and how should results be interpreted?
>
> The purpose of the three experiments is to (1) show how training models on ambiguous images can improve their accuracy when classifying other ambiguous images, (2) illustrate how DAI can be used to assess existing situation models’ performance on varying degrees of noisy data, and (3) explore how DAI can be used to directly evaluate model calibration techniques by comparing model confidence scores to DAI’s human labels. We are revising Section 5 to better convey these main points and to address related clarity issues.
>
> > The experiments do not necessarily indicate that training models on ambiguous data results in more robust models, given that studies already show that soft labels in training can improve robustness. They also do not indicate that training on ambiguous data allows models to better approximate human uncertainty.
>
> Firstly, we do not train our models using the human uncertainty judgments – we exclusively use the ambiguous images and their corresponding ground truth labels. Therefore, the experiment in Section 5.2 shows that training on ambiguous images taken directly from event-related videos can improve performance on other ambiguous images, which is a separate result from performance improvements using soft labels. Furthermore, we do not make the claim that training on this data produces models that align more closely with human uncertainty judgments. In the experiment in Section 5.3, we show that our dataset can be used as a diagnostic tool to directly evaluate model calibration. We are revising Section 5 to better clarify the significance of these experiments.
>
> > “How do you interpret the smaller HUJ MSE, when the model is trained on standard data?”
>
> We hypothesize that the standard model is more easily calibrated than the model trained on DAI because it initially has higher certainty and so calibration techniques can have a larger effect.

---

> > ### Author Response · Authors · 2022-08-28
> > **Further clarifications or questions**
> >
> > Thank you again for your review. As mentioned in the latest general comment, a new paper revision has been uploaded that addresses your comments regarding the experiments section. As the rebuttal deadline is approaching, please let us know if you have any further questions or concerns. Thanks!

---

> > > ### Comment · Reviewer_vrs4 · 2022-08-28
> > > **Clearly improved paper**
> > >
> > > Thank you for the clarifications and the revised version of your paper which addresses my mentioned concerns. Overall, it is a huge improvement to me, so that I raised my rating to 7.

---

### Official Review · Reviewer_VgV7 · 2022-07-25
**Useful dataset for robust visual event classification**

**Rating:** 7
**Confidence:** 3
**Correctness:** The construction design and implement…
**Clarity:** The paper is well-written and easy to…

**Strengths:**

DAI is carefully constructed following a strict process. It enriches the datasets for the visual event classification task.

DAI will likely be used by the community as a real-world test set to (1) assess the model's robustness/generalization ability; (2) improve existing models or develop better models; (3) develop new model uncertainty quantification approaches.



**Weaknesses:**

The presented dataset is not yet publicly accessible.

In Section 4.2, The difference between **background knowledge** and **uncertainty quantification strategies** is not clear. It seems that they are mutually inclusive.

The experiments are extensive but the results discussion can be improved:
1. In Section 5.1, when presenting the results, will it be better by showing top-10 to low human judgments? e.g., for 0-20%, 20-40%, show the top-10 performance.

2. In Section 5.2, Table 2 shows that by augmentation, the model (RN+AUG) can achieve better results on DAI than simply using the standard dataset (RN+SD). Although the performance on DAI is worse than RN+DAI, the performance on the standard test set is better. Thus, I am wondering if increasing the diversity of augmentation methods can further improve the model's performance on DAI and reach a similar level as RN+DAI.


**Additional Feedback:**

Please refer to the **Weaknesses** and **Relation To Prior Work** sections.

**Documentation:**

The benchmark dataset is well documented.

**Ethics:**

Well claimed in the paper.

**Relation To Prior Work:**

The discussion of related work is good, but I have one doubt: is there no benchmark dataset of ambiguous data for event classification except the medical images? From my understanding of Section 2.1, DAI is the first dataset for this field.



**Summary And Contributions:**

Focusing on the visual event classification task, this paper presents a benchmark dataset DAI to improve the robustness of existing models against ambiguous event images. The images are scraped from YouTube videos and annotated by carefully selected annotators, which is an admirable work. In the experiments section, the authors provide three use cases of DAI.

---

> ### Author Response · Authors · 2022-08-18
> **Thank you for your review. Regarding annotation disagreement, experiment design, and other clarifications:**
>
> Thank you for your detailed review. We have addressed common concerns in the general comment, and we are updating the paper to incorporate your feedback and revise areas that lacked clarity. Below, we have responded to your questions and concerns:
>
> > “In Section 4.2, The difference between background knowledge and uncertainty quantification strategies is not clear. It seems that they are mutually inclusive.”
>
> Based on work by Tversky and Kahneman et al. [1, 2], human probability calculations suffer from a variety of imperfections including heuristics and psychological biases. They show that these factors are not all necessarily influenced by external factors such as visual input or background knowledge, but are intrinsic cognitive mechanisms. We consider this type of noise that is not affected by such external factors as a third source of annotator discrepancies. We will update the text in Section 4.2 to clarify this.
>
> > “In Section 5.1, when presenting the results, will it be better by showing top-10 to low human judgments? e.g., for 0-20%, 20-40%, show the top-10 performance.”
>
> Yes, we agree that this metric would be more appropriate for Section 5.1. We are re-running these experiments to retrieve the top-10 values and will add them to the paper in the revision.
>
> > Based on the experiment in Section 5.2, it would be informative to include comparisons to additional augmentation techniques.
>
> Thank you for the feedback. We are running experiments that compare the model trained on DAI against a wider variety of augmentation techniques and different amounts of augmentation. Details regarding this experiment and corresponding results will be added to the next revision.
>
> > Are there any other related datasets to DAI, specifically, consisting of ambiguous data?
>
> To our knowledge, this is the first ambiguous dataset for event classification. Our novel contributions with respect to ambiguous datasets is (1) our dataset consists of intentionally ambiguous images and (2) we provide ambiguous images labeled with quantitative human uncertainty scores. We have been directed to a dataset of CIFAR-10 images labeled with distributions of human labels (CIFAR-10H), which is the most similar dataset we have seen. However, the images in CIFAR-10H depict single objects where the primary source of ambiguity is the images’ low resolution, and the human annotations are standard label classifications as opposed to quantitative uncertainty judgments. This dataset and how it differs from ours will be added to the related work section.
>
>
> [1] Tversky, Amos, and Daniel Kahneman. "Judgment under Uncertainty: Heuristics and Biases: Biases in judgments reveal some heuristics of thinking under uncertainty." science 185, no. 4157 (1974): 1124-1131.
>
> [2] Kahneman, Daniel, Olivier Sibony, and Cass R. Sunstein. Noise: A flaw in human judgment. Little, Brown, 2021.

---

> > ### Comment · Reviewer_VgV7 · 2022-08-26
> > **Thank you for your response.**
> >
> > Thank you for your answer and the revised paper. My doubts have been cleared and I will raise my rating from 6 to 7.

---

### Official Review · Reviewer_jezS · 2022-07-26
**Promising dataset of ambiguous images but with remaining questions**

**Rating:** 7
**Confidence:** 4
**Correctness:** It seems that the dataset was constru…

**Strengths:**

- The dataset seems a valuable resource for the situation/image recognition domain.

- The dataset construction seems well thought through.

- The human-perception-centric perspective is a useful angle that allows for the exploration of new questions in the field.

**Weaknesses:**

I find that the paper is sometimes not very clearly written. I'm listing my concerns here and I'm happy to increase my rating once they're clarified. [EDIT: I increased the paper score from 5 to 7 since the authors addressed most of my concerns in the author discussion here and I believe that the general proposed changes to the paper will improve its overall quality.]

- Overall, I found the verb/event terminology to be slightly confusing. The work seems motivated through situation recognition which the paper describes as "identify[ing] the verb and corresponding semantic roles (e.g. subject, object, place, reason, etc.)". If I understand correctly, the human subjects were asked for their uncertainty judgments based on the event type (e.g., "wedding") whereas model uncertainty was established based on verbs. If so, it's not clear to me to what extent they can be compared. It would be useful to include a clarifying sentence on the verb/event terminology used and how each of them is used in the experiments and argument of the paper.

- While I enjoyed the analysis of the sources of annotator disagreement (Section 4.2), it seems that the core sub-dataset remains underexplored. Given the aim of the paper to have a dataset of ambiguous images, it seems that images that are rated at around 50% confidence with high agreement comprise the actual dataset that falls under that definition. Was there any specific analysis done on the high-agreement data? The 50% bin is represented in Table 1 and 2 but did specifically the ambiguous data in those middle bins have any effect on the results reported in Table 3? Or is it simply the noisier data that causes the results?

- I'm not quite sure about the goal of the experiment in Table 2. If the goal is to achieve higher accuracy on the verb classification across confidence bins, training on DAI data is clearly beneficial. However, I'm unsure whether another goal was to have a model that is similarly calibrated to human certainty in which case the lowest confidence bin is not well calibrated. A discussion of this would be useful.

- In Section 4.1, the authors use Spearman correlation for analyzing inter-annotator agreement. Since other metrics for inter-annotator agreement such as Fleiss' Kappa or Krippendorff's alpha are specifically designed for such goals, it would be great to include an argument why Spearman correlation was used instead.

- The related works section is not quite clear yet on how the presented work is similar and different from previous work. While it repeatedly states that other work is similar and this paper is different, how exactly it is similar and different would be useful to include. (Specifically in section 2.2 but it's a general recommendation for all of section 2.)

**Additional Feedback:**

The figure caption for Figure 5 would benefit from a short summary of what it's showing (for instance "Examples for potential sources of annotator disagreements").

**Clarity:**

Overall, the paper has a structure that can be easily followed and is writing style is very clear. However, as mentioned in the weaknesses, I still have some remaining questions which I hope can be clarified in the paper as well.

**Documentation:**

There seems sufficient detail provided for reproducibility.

**Ethics:**

The authors highlight ethical considerations. As proposed by the authors, it seems sensible to ensure that data doesn't stay in the dataset if the original video was deleted by the original owner.

**Relation To Prior Work:**

I think the paper could be more clear about in what ways it comes apart from specific work mentioned in the related works section.

**Summary And Contributions:**

This paper introduces the DAI dataset, a Dataset of Ambiguous Images. While previous work tends to focus on creating datasets with images that are a challenge to models, this work focuses on images that are challenging for people and provides an opportunity for investigating model capabilities on this cognitively interesting set. The dataset is comprised of images extracted from youtube videos so that they capture a wide range of salience and noise, and were selected based on predefined event types. The authors then compare the confidence scores of humans and models as to whether the image represents a certain event, providing a basis for future model evaluations. They also find that training models based on more ambiguous data increases their accuracy, underlining the value of such noisy image datasets.

---

> ### Author Response · Authors · 2022-08-18
> **Thank you for your review. Regarding verb classification, novel contributions, and other clarifications:**
>
> Thank you for the thoughtful comments and corrections. We are revising the paper to address reviewers’ concerns, and we hopefully clarify your specific questions below. We have additionally addressed common concerns from reviewers in the general comment.
>
> > What is the distinction between situation recognition, event classification, and verb classification? Why do you use verb classification in experiments?
>
> Situation recognition decomposes events into two parts: The verb, which dictates the structure of the event, and semantic roles, which determine the details of the event. Since it determines the structure of the depicted event, we feel that verb prediction serves as the foundational task for situation recognition. In this paper we focus on characterizing the ambiguity associated with the general event type and structure as opposed to the ambiguity of more specific event-centric attributes, and so this is what we target in our experiments.
>
> > How does this paper differ from past work, especially from papers mentioned in Section 2.2?
>
> The key aspects of our work that distinguish it from prior research are (1) our dataset consists of intentionally ambiguous images depicting semantically complex situations, and (2) we use quantitative human uncertainty judgments as labels for this image dataset. In Section 2.2, the first paragraph primarily covers cognitive science research that considers human uncertainty from a theoretical perspective. Then, we cite Misra et al. [1], Chen et al. [2], and Uma et al. [3], who cover human uncertainty in the context of machine learning. Misra et al. and Uma et al. explore disagreement in human-annotated discrete labels for images, whereas in this paper we explicitly collect and analyze individual quantitative uncertainty judgment. This is similar to Chen et al.’s annotation process, but their uncertainty judgments are for text entailment questions whereas ours correspond to ambiguous images.
>
> > Why don’t we focus on the images that elicit high-agreement annotations that average around 50% certainty?
>
> Our aim in this paper is to explore the full range of visual input associated with a given event type, even when such images elicit disagreement or average scores above or below 50%.  Therefore, we believe that these other images that don’t score near 50% are an important part of the dataset, even if they aren’t optimally uncertain by some definitions. Focusing an analysis on the most clearly ambiguous images in the dataset is an interesting direction that has useful applications, which we leave to future work.
>
> > It isn’t clear what the purpose of the experiment in Section 5.2 is, and an explanation for the higher performance + poorer calibration of the RN+DAI model would help readers to interpret the results.
>
> Yes, the goal of the experiment in Section 5.2 is to illustrate that better accuracy on ambiguous images can be achieved by models trained on ambiguous data. However, we agree that the poorer calibration of this model is an interesting result worth discussing. While it may partially be due to poorer calibration in general, we hypothesize that RN+DAI has unusually high performance on these lower bins at least partially because of the small set of classification labels in the experiment. Human annotators were not provided with a list of possible event types, while the models in 5.2 only had to choose between 4 events (we are constrained to these events by the available human labeled DAI data and publicly available high-certainty images). Therefore, all three models likely have much higher accuracy than if they were considering a larger set of events like humans were. We will update the results analysis in Section 5.2 to address this.
>
> > Why was Spearman correlation used to analyze annotation agreement instead of a metric such as Fleiss’ Kappa or Krippendorff’s alpha?
>
> Historically, Spearman correlation has often been used for measuring agreement for scalar annotations (“VideoMem” by Cohendet et al. [4], “Efficient Online Scalar Annotation with Bounded Support” by Sakaguchi et al. [5], “Quality Assessment for Crowdsourced Object Annotations” by Vittayakorn et al. [6], etc.). However, it would be interesting to compare this against another metric, and so we are running data analysis using alternate metrics. This analysis will be added to the next revision.
>
> (Citations in second comment)

---

> > ### Author Response · Authors · 2022-08-18
> > **Citations**
> >
> > [1] Misra, Ishan, C. Lawrence Zitnick, Margaret Mitchell, and Ross Girshick. "Seeing through the human reporting bias: Visual classifiers from noisy human-centric labels." In Proceedings of the IEEE conference on computer vision and pattern recognition, pp. 2930-2939. 2016.
> >
> > [2] Chen, Tongfei, Zhengping Jiang, Adam Poliak, Keisuke Sakaguchi, and Benjamin Van Durme. "Uncertain natural language inference." arXiv preprint arXiv:1909.03042 (2019).
> >
> > [3] Uma, Alexandra N., Tommaso Fornaciari, Dirk Hovy, Silviu Paun, Barbara Plank, and Massimo Poesio. "Learning from disagreement: A survey." Journal of Artificial Intelligence Research 72 (2021): 1385-1470.
> >
> > [4] Cohendet, Romain, Claire-Hélène Demarty, Ngoc QK Duong, and Martin Engilberge. "VideoMem: Constructing, analyzing, predicting short-term and long-term video memorability." In Proceedings of the IEEE/CVF International Conference on Computer Vision, pp. 2531-2540. 2019.
> >
> > [5] Sakaguchi, Keisuke, and Benjamin Van Durme. "Efficient online scalar annotation with bounded support." arXiv preprint arXiv:1806.01170 (2018).
> >
> > [6] Vittayakorn, Sirion, and James Hays. "Quality Assessment for Crowdsourced Object Annotations." In BMVC, pp. 1-11. 2011.

---

> > ### Comment · Reviewer_jezS · 2022-08-18
> > **Thank you for the clarifications**
> >
> > Thank you for the detailed and thoughtful responses. Your clarifications have been very helpful and I especially appreciate the discussion on calibration.
> >
> > I just wanted to confirm that in the human subject experiments as well as in the model prediction task, the investigated labels are instances like "wedding" or "birthday party" and not verbs, right? From the paper, it's clear to me that these are the labels for the human subject experiments but I just want to make sure that that's the same for the model.
> >
> > Once this is clarified, I'll increase my score.
> >
> > I do still think that a dataset that is specifically about ambiguous images that's motivated from a cognitive perspective would be even stronger if it elaborated on the way images can be ambiguous (e.g., images with disagreement among raters but where each judgment has high confidence vs. images with judgments that have high individual uncertainty vs. images that people seem to agree on with high confidence). However, I understand the argument the authors make and don't see this as a reason for rejecting this clearly interesting resource.

---

> > > ### Author Response · Authors · 2022-08-18
> > > **Clarification of labels**
> > >
> > > Thank you for your quick response. Yes, the labels used in the human subject experiments and the model prediction tasks in Section 5.2 and 5.3 are event instances like “wedding”, “birthday party”, etcetera. Verbs are only used as labels for the experiment in Section 5.1 to align with the labels SoA situation recognition models are trained on.
> > >
> > > We agree that a further analysis of the dataset as you described would be very interesting future work, and we will note this in the revision.

---

> > > > ### Comment · Reviewer_jezS · 2022-08-19
> > > > **Increased score**
> > > >
> > > > Thank you for the clarification -- that makes sense to me now. I would recommend communicating this clearly in the paper to avoid any confusion.
> > > >
> > > > Since my main concerns are addressed, I've increased my score accordingly. Thank you for engaging with all of my points!

---

### Official Review · Reviewer_UAMr · 2022-07-28
**Useful dataset, smart approach**

**Rating:** 8
**Confidence:** 4

**Strengths:**

1. Strong motivation: Ambiguous images are very common in the real world and so it is important to include them while testing models to obtain a more accurate measure of real-world performance. Moreover, most existing datasets are simple for humans and therefore do not capture the complete range for human performance. Harder datasets allow for a fairer comparison between models and humans. The dataset proposed in this paper contributes effectively towards both these issues.
2. Extracting images from videos to increase ambiguity and reduce bias towards saliency is a very smart idea.
3. Methods used in this paper are systematic and rigorous. From considering 2 variations of the uncertainty rating task to filtering out bad annotators to using FrameNet templates to measure semantic similarity, the authors do a good job of being objective in their experiments.

**Weaknesses:**

The task used to collect data varies a lot between experiments in the paper. The DAI dataset is collected on 20 event types. Human data is collected on only a subset of 6 event types. Humans are asked to report uncertainty scores given ground-truth label while networks, in all but the last experiment, are evaluated on a verb prediction task, assuming that it is completely indicative of situation recognition. Why not just train networks on situation prediction. Training paradigm assessment uses a subset of only 4 event types. Uncertainty quantification task on models is done using a binary classification task. All these variations make it very difficult to interpret results and makes one wonder if say, verb prediction accuracy is indeed a suitable for comparison with human uncertainty ratings. Why not train models on a situation prediction task? It would help if the authors would provide justification for why such subsets were selected and why nature of tasks were so varied.

**Additional Feedback:**

I have some questions which will help me make a more informed decision:
1. Why is the ResNet50 idea only to select frames for short videos? Why not for all? For long, relatively static videos, how do you ensure diversity in images?
2. Were subjects allowed unlimited response time?
3. Why is RN+SD (trained on standard dataset) so much better than JSL (trained on SWiG) if both were trained on high-certainty images?

**Clarity:**

The paper is well-written. Figures and captions are useful to understand motivation, experiments and results.

**Correctness:**

All aspects of experimental design and data collection are sound, to the best of my knowledge.

**Documentation:**

Documentation of dataset and code for models is provided in supplementary material.

**Ethics:**

Data collection methods are ethical to the best of my knowledge.

**Relation To Prior Work:**

Relevant prior work has been clearly outlined in the Related Work section.

**Summary And Contributions:**

Given that most popular vision datasets are easy for humans, this paper proposes a method to collect ambiguous data by extracting images from videos. Using this method, the authors collect a sizeable dataset, and subjective human event recognition uncertainty scores on a subset of it. They find that SOTA situation recognition models perform poorly on images where humans are <80% confident. Training on ambiguous data alleviates this problem. Finally, using model calibration approaches, they achieve low MSE between model and human uncertainties on a binary event classification task, relative to the baseline.

---

> ### Author Response · Authors · 2022-08-18
> **Thank you for your review. Regarding experiment design and other clarifications:**
>
> Thank you for the thoughtful review. We are in the process of revising the paper, which will be uploaded before the rebuttal deadline. We have addressed common concerns in the general comment and we aim to clarify your specific questions below:
>
> > Why is a verb prediction task used when the paper considers event classification/situation recognition?
>
> Verb prediction serves as the foundational task for situation recognition, as it determines the structure of the depicted event (while the full situation recognition task also considers details, or semantic roles, such as “who”, “where”, etc.). In this paper we focus on characterizing the ambiguity associated with the general event type and structure as opposed to the ambiguity of more specific event-centric attributes, and so this is what we target in our experiments.
>
> > Humans report uncertainty scores given the ground-truth label.
>
> This is only the case for Task A. For Task B, there is a 50% chance that the event type prompt is any one image’s ground truth event label. Furthermore, for both tasks, annotators are not told what the actual ground truth of any image is. They are only given an event type prompt that may or may not be a correct label for any one image.
>
> > “Human data is collected on only a subset of 6 event types”
>
> Collecting a sufficient number of annotations for 600 images is fairly costly, and it was important to us to have all 600 images annotated for any annotated event to produce a sufficiently representative uncertainty distribution for the event type. Therefore, we elected to limit the number of event types with human labels to keep costs reasonable while focusing on our highest priorities for the dataset.
>
> > Why are 4 classes used for the experiment in Section 5.2? Why is a binary classification task used for the experiment in Section 5.3?
>
> We train the models in Section 5.2 on 4 events because these are the four event types in DAI that both have human judgment scores and are sufficiently represented in high-certainty image datasets. While we use the same models and training datasets in Section 5.3, we specifically use a binary classification task for this experiment to match the human annotation process since we make a direct comparison between the model confidence scores and human labels.
>
> > “Why is the ResNet50 idea only to select frames for short videos? Why not for all? For long, relatively static videos, how do you ensure diversity in images?”
>
> We did apply the clustering algorithm to each video’s extracted frames regardless of length. We removed relatively static videos during the manual curation process detailed in Section 3.1. These points were not made sufficiently clear in the paper, and will be clarified in the revision.
>
> > Were annotators allowed unlimited time?
>
> Annotators were given a 24-hour window to submit any one set of annotations, which are estimated to take under 1 minute. This information will be added to the appendix.
>
> > “Why is RN+SD (trained on standard dataset) so much better than JSL (trained on SWiG) if both were trained on high-certainty images?”
>
> RN+SD is trained to predict between 4 classes, while the JSL verb classifier is trained to predict between 504 classes. This is noted on lines 263-264 in the submission. With 504 classes, there is a much higher likelihood that, for any given image, there are multiple classes that at least partially align with the image’s contents.

---

> > ### Comment · Reviewer_UAMr · 2022-08-22
> > **Thank you for the clarification!**
> >
> > Thank you for responding to all my questions. Methods used in the paper are much clearer to me now.

---

### Author Response · Authors · 2022-08-18
**General comment**

We thank the reviewers for their thoughtful responses to our paper. We noticed a few common concerns that we would like to address here. We have also provided individual responses to each of the reviewers and are currently updating the manuscript.

**We would like to elaborate on the novel contributions of our work (Reviewers jezS, eRWV, and gzHA).** We want to emphasize that we present the first dataset of **intentionally ambiguous event-centric images**. Our contribution can be summarized from two aspects.

Firstly, we introduce a method to collect images that mitigates the reporting bias found in existing image datasets to model the distribution of visual input experienced by humans. We argue that having datasets that more closely model this distribution is important for developing models that are robust to the range of visual data present in many real life scenarios, such as human-robot collaboration, and to our knowledge this is the first dataset that attempts this.

Furthermore, to our knowledge this is the first event-centric image dataset that uses **quantitative human uncertainty judgments as labels**. We show in the experiments that these labels can be used to assess model accuracy on different distributions of ambiguous data and additionally directly assess model calibration techniques.

**Why are we focusing on verb classification for the experiments (Reviewers UAMr and jezS)?**

We focus on verb prediction because it is the foundational task of situation recognition that the rest (semantic role prediction) builds off of. In situation recognition, events are broken down into (1) a verb (e.g. “jumping”) and (2) a set of semantic roles dependent on that verb (e.g. agent: “boy”, source: “rock”, destination: “water”). Most approaches to situation recognition first identify the verb in an image, and then pass that verb into a second module that identifies the semantic roles. So, the verb dictates the rest of the classification and the model’s performance is upper bounded by its verb accuracy. In this paper, we are interested in the ambiguity associated with the verb, or foundation of situation recognition, as opposed to ambiguity associated with the details.

**Regarding our experiment setup (Reviewers UAMr, jezS, vrs4, and eRWV)**: We first clarify the purposes of the three experiments: (1) Illustrating how DAI can be used to assess existing situation models’ performance on varying degrees of noisy data, (2) showing how training models on ambiguous images can improve their accuracy when classifying other ambiguous images, and (3) exploring how DAI can be used to directly evaluate model calibration techniques by comparing model confidence scores to DAI’s human labels.

To achieve (1), we evaluate three state-of-the-art situation recognition models on a verb classification task between the two event types. We choose the task such that the two event types both have human annotations and also align with the traditional situation recognition ontology.

For (2), we train one ResNet model on 4 event types using DAI (with ground-truth labels), and another two ResNet models on those same 4 event types using “high-certainty data” (with and without augmentation). The event types we used are the only events with human annotations that are also sufficiently represented in high-certainty image datasets. We compare these three models on a held-out validation set of DAI images.

Finally, to show (3) we evaluate a set of uncertainty quantification approaches using the ResNet models described in experiment (2) by calculating the MSE between the model confidence scores and the human judgment annotations for each image. For this experiment, we train and evaluate the models on a set of binary classification tasks using the data and event types described in experiment (2) to match the human annotation task.

We provide further clarifications to individual comments in the replies, and will improve the clarity of the experiments section in the revised paper.

---

### Author Response · Authors · 2022-08-25
**New revision uploaded**

Thank you to all the reviewers for providing such detailed feedback. We have uploaded a revised version of the paper that incorporates this feedback. Below is an overview of the changes made.

- **Section 1 + Section 2**: Novel contributions and differences from past work have been emphasized.
- **Section 2**: Additional related works have been added.
- **Section 3, Appendix A.1, Appendix A.2**: Minor clarifications about dataset creation have been added.
- **Section 4.2**: Analysis of intra-annotator agreement has been added.
- **Section 4.3**: The “Uncertainty quantification strategies” passage has been revised to better differentiate it from other sources of inter-annotator variance.
- **Section 5**: The section has been rewritten to improve clarity. Experiments are now organized into “Models”, “Task”, and “Results” subsections, experiments have been reordered, an overview has been added to the beginning of the section, and results have been rewritten to emphasize key takeaways.
- **Section 5**: Added additional augmentation methods to the “Training on DAI for Event Classification” experiment.
- **Section 5**: Top-5 accuracy metric was replaced with top-10 accuracy in the “Evaluating Verb Prediction Models” experiment.
- **Section 5**: An ECE metric has been added to the “Evaluating Uncertainty Quantification Methods” experiment.
- **Section 5**: “Training on DAI for Event Classification” and “Evaluating Uncertainty Quantification Methods” experiments have been run across multiple seeds and standard deviation is now reported.
- **Section 5, Appendix D.2**: Text explaining why verb classification is used for evaluating situation recognition models has been added.
- **Section 6.3**: A discussion considering alternate approaches to dataset creation and analysis has been added.
- **Appendix C**: An analysis comparing agreement metrics (Spearman, Fleiss’s kappa, Krippendorff’s alpha) for the human annotations has been added.
- **Appendix D**: Additional experiment details have been added (which annotations were used for which experiments, augmentation filters, etc.)
- **Appendix D.3**: An experiment comparing the mean as an aggregation metric for binning human annotations to an alternate binning method has been added.
- **Appendix D.4**: HUJ MSE is compared against KL divergence for measuring model calibration in the  “Evaluating Uncertainty Quantification Methods” experiment.
- Minor errors have been corrected and clarifications have been added (Fig 5 caption, S4 inter-annotator variance discussion)
- **Supplementary materials**: Code has been updated to include new experiment setups, and the experiment setup README has been extended.

---

### Meta-Review · Area_Chair_KNMD · 2022-09-07

**Recommendation:** Accept
**Confidence:** 4

**Metareview:**

Overview:
Accept (poster)

The reviews for “Ambiguous Images With Human Judgments for Robust Visual Event Classification” were originally somewhat mixed, but converged to mostly positive reviews of the revised text / clarifications. Many of the originally noted issues had more to do with the clarity of presentation than issues with the underlying work and with the final revision the outstanding issues were cleared up.

Strengths:
* Important topic and rarely considered. This work makes a strong contribution in the form of a dataset of ambiguous images along with human uncertainty annotations to quantify the ambiguity.
* Clever method of obtaining ambiguous data from video.
* Clear and rigorous annotation procedures

Weaknesses:
* While this provides a clear step forward in evaluating models with-respect-to ambiguous data, it doesn’t provide broadly applicable solutions (focused on event detection, limited number of classes, and training on ambiguous data presents a trade-off with accuracy on standard data). That being said, this is still a clear step forward in an under-considered area and was a well-thought out contribution.


Contributions:
The paper describes a dataset and evaluations which are original in the following ways: First, it contains images which are ambiguous with respect to the event they depict, while most existing computer vision datasets focus on unambiguous images / data. Secondly, they quantify the ambiguity / uncertainty using human raters and human uncertainty judgements. Finally, they show how standard models perform when trained using standard data and the provided ambiguous data and demonstrate improvements in ambiguous scenarios over standard datasets.

Significance:
Ambiguous scenarios are clearly common and relevant in a variety of real world tasks and it is important to be able to evaluate how existing models do in these scenarios. This also allows for better comparisons between models and humans and provides a method of evaluating the calibration of standard models. That being said, the dataset is still rather small for training and has a limited number classes, is focused on event classification, and the provided results suggest a tradeoff between performance on standard data and ambiguous data. So while this work does provide a clear step forward for evaluating / diagnosing how standard models behave with ambiguous images / calibration, it doesn’t provide broadly generalizable solutions.

Quality:
The data collection is well thought out (using video data for overall clear scenarios, but ambiguous images is a clever idea) and well motivated. The model experiments provide a clear motivation for collecting additional ambiguous data over standard data augmentation techniques and again highlights the importance of calibrating ML models.

Clarity:
The initial paper was challenging to follow, but the reorganization and clarifications provided in the revision were a significant improvement. It’s now much easier to follow contributions and experiments.

---

### Decision · Program_Chairs · 2022-09-16

Accept